# Fully autonomous mouse behavioral and optogenetic experiments in home-cage

Yaoyao Hao, Alyse Marian Thomas, Nuo Li*

Department of Neuroscience, Baylor College of Medicine, Houston, United States

**Abstract** Goal-directed behaviors involve distributed brain networks. The small size of the mouse brain makes it amenable to manipulations of neural activity dispersed across brain areas, but existing optogenetic methods serially test a few brain regions at a time, which slows comprehensive mapping of distributed networks. Laborious operant conditioning training required for most experimental paradigms exacerbates this bottleneck. We present an autonomous workflow to survey the involvement of brain regions at scale during operant behaviors in mice. Naive mice living in a home-cage system learned voluntary head-fixation (>1 hr/day) and performed difficult decision-making tasks, including contingency reversals, for 2 months without human supervision. We incorporated an optogenetic approach to manipulate activity in deep brain regions through intact skull during home-cage behavior. To demonstrate the utility of this approach, we tested dozens of mice in parallel unsupervised optogenetic experiments, revealing multiple regions in cortex, striatum, and superior colliculus involved in tactile decision-making.

## Introduction

Goal-directed behavior is orchestrated by activity distributed across multiple brain regions. A starting point for understanding how distributed activity mediates a single behavior is to identify activity that causally contributes to the behavior. For example during perceptual decisions, activities that correlate with sensation, choice, and movement are distributed across distinct brain areas (*Gold and Shadlen, 2001*; *Hernández et al., 2010*; *Guo et al., 2014a*; *Siegel et al., 2015*; *Sippy et al., 2015*; *Brody and Hanks, 2016*; *Svoboda and Li, 2018*; *Allen et al., 2019*; *Crochet et al., 2019*; *Mayrhofer et al., 2019*; *Pinto et al., 2019*; *Steinmetz et al., 2019*; *Li and Mrsic-Flogel, 2020*). Delineating which activity casually contributes to decision-making requires spatially and temporally precise manipulation of specific activity that is widely dispersed across the brain.

The mouse is particularly suitable for comprehensive analysis of neural activity due to the small size of the brain. Modern optogenetic methods can manipulate activity in specific brain regions with excellent temporal resolution (*Deisseroth, 2015*; *Wiegert et al., 2017*; *Li et al., 2019*), but optogenetic experiments currently can only probe a limited number of brain regions in single studies. In standard optogenetic experiments, mice are trained in operant behavior and optogenetic testing is carried out in daily sessions to manipulate individual brain regions. This process is serial and slow, prohibiting comprehensive surveys of many brain regions during complex behaviors.

One bottleneck results from manual operant conditioning training, which is required in most experimental paradigms. For example, training mice in decision-making tasks requires significant human involvement in evaluating mice performance and modifying task parameters to gradually shape behavior toward high performance (*Guo et al., 2014b*; *Burgess et al., 2017*; *Aguillon-Rodriguez et al., 2020*). This process is laborious and requires human expertise. Such expertise is difficult to transfer across experimenters and across labs. The low throughput also rises significant barriers for explorations of more complex decision-making tasks, due to the significant time and effort required to explore many task parameter variations. The other bottleneck is due to the serial nature of optogenetic testing. In particular, existing optogenetic methods probe deep brain regions

*For correspondence:
nuol@bcm.edu

**Competing interests:** The authors declare that no competing interests exist.

using optical fibers, which target one brain region at a time, are labor-intensive to implant, and require manual tethering of light source to the fiber implant. An experimental framework to swiftly survey the behavioral involvement of many brain regions at scale would significantly speed up mapping of brain networks contributing to decision-making or other goal-directed behaviors.

Automated experiment can potentially overcome these bottlenecks. Automated systems can train rodents in behavioral tasks by changing task parameters based on performance free of human supervision, thus enabling parallel and high-throughput experiments (*Kampff et al., 2010*; *Erlich et al., 2011*; *Poddar et al., 2013*; *Scott et al., 2013*; *Murphy et al., 2016*; *Aoki et al., 2017*; *Bollu et al., 2019*; *Erskine et al., 2019*; *Qiao et al., 2019*; *Aguillon-Rodriguez et al., 2020*; *Bernhard et al., 2020*; *Murphy et al., 2020*). Moreover, automated training provides standardization that frees the training process from idiosyncratic human interventions and documents the entire training process. Automated training has been extended to train rodents in home-cages (*Poddar et al., 2013*; *Aoki et al., 2017*; *Silasi et al., 2018*; *Bollu et al., 2019*; *Erskine et al., 2019*; *Qiao et al., 2019*; *Bernhard et al., 2020*; *Murphy et al., 2020*), opening the possibility of prolonged behavioral training that permits more difficult decision-making tasks. In some cases, automated systems can also be incorporated into a large environment to probe effects of social and environmental factors on cognitive behaviors (*Freund et al., 2013*; *Castelhano-Carlos et al., 2014*; *Torquet et al., 2018*).

However, significant aspects of home-cage training still need to be improved and validated to enable high-throughput experiments. First, it remains to be determined whether mice can robustly learn challenging decision-making tasks under home-cage operant conditioning. Existing home-cage trainings are limited to relatively simple behavioral tasks and modest training durations. Second, it remains to be determined whether behaviors resulting from home-cage training resemble human-supervised training and whether they engage the same brain areas. For example, cortical regions contributing to perceptual decisions can vary across tasks and training conditions (*Chowdhury and DeAngelis, 2008*; *Licata et al., 2017*; *Liu and Pack, 2017*; *Gilad et al., 2018*; *Hong et al., 2018*). Finally, home-cage training has not been integrated with unsupervised optogenetic testing. Automation could potentially enable comprehensive optogenetic experiments targeting many brain regions during complex behaviors.

Here, we introduce a fully autonomous workflow that combines home-cage behavioral training and optogenetic testing. We introduce a low-cost standalone home-cage system that allows robust training in difficult decision-making tasks. Completely naive mice self-engaged in prolonged voluntary head-fixation (>1 hr/day) and underwent continuous training and testing for 2 months without human supervision. In the context of automated home-cage behavior, we integrated a fiber-free optogenetic method to manipulate cortical and subcortical regions through an intact clear skull. Electrophysiological recordings show that photostimulation could potently modulate neural activity in deep brain structures such as the striatum and midbrain. We collected an extensive benchmark dataset (113 mice, 1.92 million trials) training mice in a tactile decision task with a short-term memory component to show that mice in automated training learned the task using similar behavioral strategies as mice in manual training. Optogenetic loss-of-function experiments show that the learned behavior engaged the same cortical regions. The hardware design files, software, and task training protocols for the home-cage system are made publicly available along with extensive documentations for other researchers to implement similar automated training for other operant behaviors.

Our automated home-cage system significantly lowers the barrier for training mice in difficult decision-making tasks. To demonstrate this utility, we show that mice could robustly learn contingency reversals in which they flexibly reported tactile decisions using directional licking, a behavior that was previously difficult to attain in manual training. In addition, our workflow is particularly suitable for mapping cortico-basal-ganglia loops involved in goal-directed behaviors. The striatum, its cortical inputs, and downstream output nuclei are topographically organized (*Hintiryan et al., 2016*; *Hunnicutt et al., 2016*; *Hooks et al., 2018*; *Peters et al., 2019*; *Lee et al., 2020*). However, a systematic survey of different striatal domains' involvement in specific behaviors has not been achieved. We demonstrate the utility of our workflow in high-throughput optogenetic mapping, revealing multiple subregions in the striatum and downstream superior colliculus critical for tactile-guided licking decisions. Our workflow opens the door to rapidly survey distributed brain networks driving goal-directed behaviors.

## Results

### Workflow overview for autonomous behavior and optogenetic experiments

Our goal is to develop an automated workflow to swiftly probe the involvement of many brain regions in a single perceptual decision task. To accomplish this, we target specific brain regions for optogenetic manipulation in individual cohorts of mice. Mice undergo standardized behavioral training in perceptual decision tasks. After training, the targeted brain regions are perturbed during specific behavioral epochs to examine their involvement in the behavior (*Figure 1A*). Across different cohorts of mice, different brain regions are tested. Two bottlenecks addressed in this workflow are manual behavioral training and manual optogenetic testing (*Figure 1B*).

To overcome these bottlenecks, we designed a robust home-cage system for mice to voluntarily engage in head-fixation that was amenable to operant conditioning and optogenetic testing (*Figure 1C*). A behavioral test chamber was built onto the mouse home-cage and ran autonomously without human supervision. Mice accessed the test chamber through a headport and engaged in behavioral tasks (*Video 1*). Automated computer algorithms trained naive mice to perform head-fixation and decision-making tasks. In the context of unsupervised behavioral testing, we integrated an optogenetic method to manipulate activity in specific brain regions. The entire process ran autonomously 24/7 for 2 months or longer (*Figure 1A–B*).

To build the behavioral test chamber, we designed a 3D-printed 'L'-shaped board which could be attached to standard mouse cages (*Figure 1C*). An opening (20 mm wide) in the center formed a headport. Mice with headbar implants enter the headport in head-restrained configuration from the home-cage (*Figure 1D*). A motorized lickport in front of the headport dispensed water reward. The lickport was actuated by two linear motors, moving the lickport toward or away from the mouse. The stimulus for the decision-making task was a mechanical pole on the right side of the headport. The pole was moved vertically by a piston to stimulate the whiskers at different locations to instruct a tactile decision (see *tactile decision task* below). The location of the pole relative to the mouse was controlled by another motor. Inside the home-cage, mice accessed the headport on an elevated platform (*Figure 1C*, inset). The platform was embedded with a micro load cell. The weight of the mouse could be read out from the load sensor, which eliminated daily human interventions to measure mouse body weight.

To make the system run standalone, microcontrollers (Arduino) were used to control the whole system (Materials and methods, *Figure 1—figure supplement 1A*). A master microcontroller controlled the progression of head-fixation training and task training. The task difficulty was gradually increased to facilitate learning. A second microcontroller was triggered by the master controller and it ran finite-state machines that controlled individual behavioral trials with high temporal precision (0.1 ms). The master controller was equipped with a SD card that stored mouse-specific metadata, task parameters, and behavioral data. Each mouse had its unique SD card and could use it to run on any home-cage system. Optionally, the system could be connected to a PC to display behavioral performance and monitor training progression in real-time (*Figure 1—figure supplement 1B*). The entire system was fit into a self-contained enclosure (*Figure 1D*, 56 × 25 × 23 cm). Multiple systems could be packed onto a standard rack in a small space to enable parallel testing (*Figure 1—figure supplement 1C*).

To screen for brain regions involved in behavior, we adapted a fiber-free optogenetic strategy that non-invasively manipulated activity in specific brain regions though an intact skull. For each mouse, we virally expressed red-shifted opsins in a targeted brain region. Mice were prepared with a clear skull implant that provided optical access to the brain (*Guo et al., 2014a*). During head-fixed behavior, 630 nm light emitted from above the headport to broadly illuminate the targeted brain region and photostimulate the locally expressed opsins (*Video 2*). Red light can penetrate deep in neural tissue (*Tromberg et al., 2000*; *Liu et al., 2015*; *Wiegert et al., 2017*; *Li et al., 2019*) and thus can non-invasively manipulate deep brain regions (*Lin et al., 2013*; *Chuong et al., 2014*; *Klapoetke et al., 2014*).

The integrated workflow thus overcame the bottlenecks of manual behavioral training and manual optogenetic testing (*Figure 1A–B*). Completely naive mice learned to perform tactile decision-making and underwent optogenetic testing in their home-cage without human supervision. A large number of brain regions can be tested in parallel across different cohorts of mice.

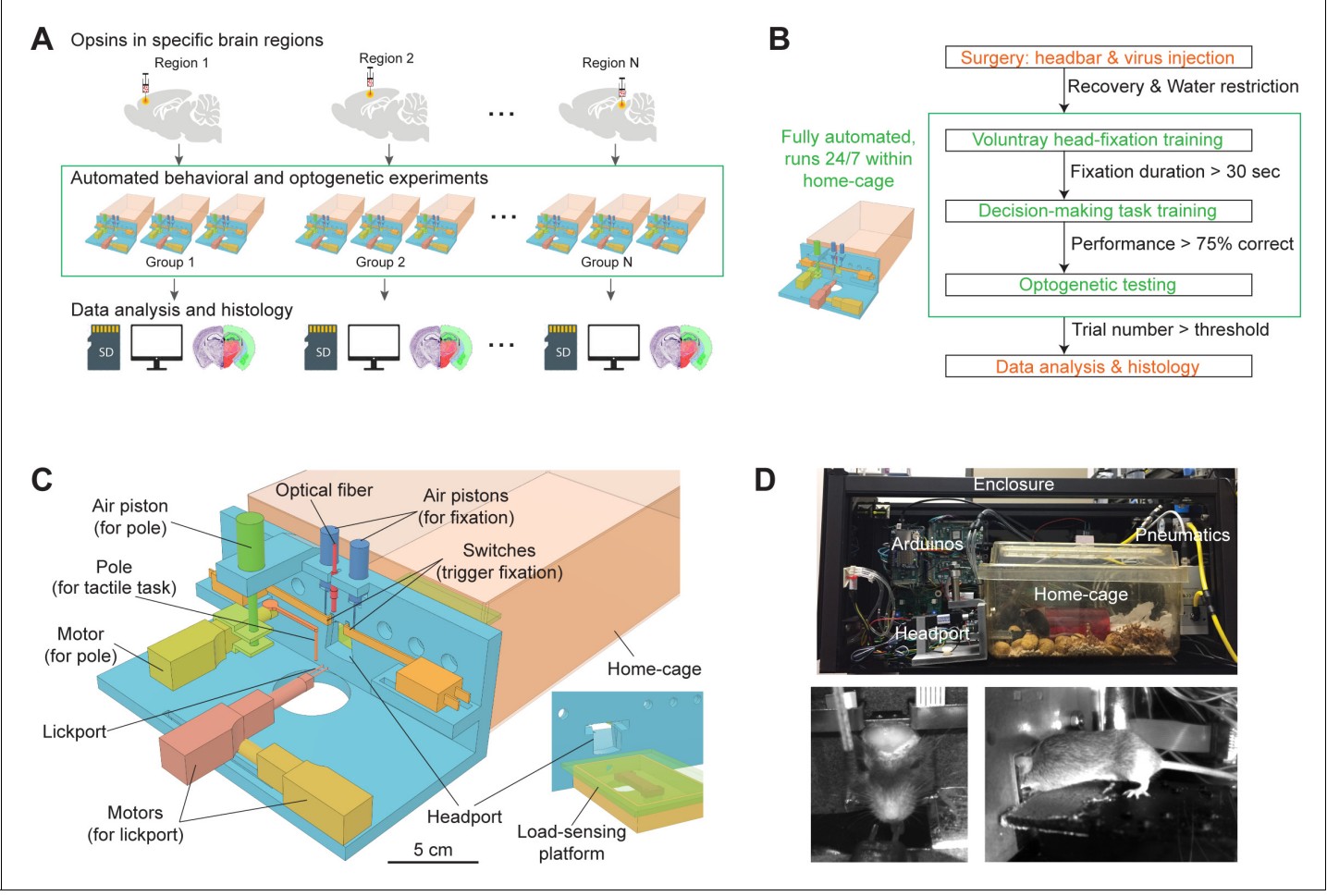

**Figure 1.** Workflow for autonomous behavior and optogenetic experiments and design of home-cage system. (**A**) Workflow for autonomous behavioral and optogenetic experiments. In each group of mice, optogenetic opsins are expressed in a specific brain region. Naive mice undergo autonomous behavioral training and optogenetic testing in their home-cage. Multiple groups of mice are tested in parallel to examine multiple brain regions. Data is stored on SD cards for analysis. Histology is performed at the end of the workflow to register the targeted brain regions to an atlas. Green bounding box highlights the portion of the workflow that is unsupervised by experimenters. (**B**) Workflow for automated behavioral training and optogenetic testing. After recovery from surgery, mice are housed in the home-cage system 24/7. Automated computer algorithms train mice to perform voluntary head-fixation, decision-making task, and carry out optogenetic testing. The progression in the workflow is based on behavioral performance. Green bounding box corresponds to the bounding box in (**A**). (**C**) Design of the home-cage system. The main component is a behavioral test chamber which can be accessed through a headport from the home-cage. Inset shows the view of the headport from inside the home-cage. Mice access the headport on a load-sensing platform. See *Figure 1—figure supplement 1* and Materialsand methods for details. (**D**) Photographs of the home-cage system. Top: side view of the system. The system is standalone with controllers (Arduinos) and actuators packed into a self-contained enclosure. Bottom, the front and back view of a mouse accessing the headport and performing the tactile decision task.

The online version of this article includes the following figure supplement(s) for figure 1:

**Figure supplement 1.** Overview of the home-cage system.

## Voluntary head-fixation in home-cage

We adapted a head-fixation mechanism that was previously designed for head immobilizations in rats (*Scott et al., 2013*). Two pneumatic pistons pressed against a custom titanium headbar to immobilize the head. The headbar (*Figure 2A*) was processed with two kinematic depressions that were fit to the cone shaped tips of the pneumatic pistons, which mechanically brought the headbar to the same position upon head-fixation. This head-fixation mechanism was integrated into the headport that accessed the behavioral test chamber (*Figure 1C*). *Figure 2B* shows the sequence of a head-fixation and release cycle. Head-fixation was triggered by mouse entry into the headport. The two wings of the headport have widened tracks to guide headbar entry. The tracks funneled to

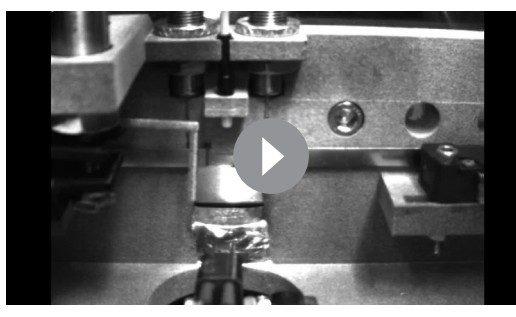

**Video 1.** A mouse performing voluntary head-fixation, tactile decision task, and self-release in home-cage. A mouse voluntarily pokes into the headport and gets head-fixed. Once head-fixed, a trial is initiated. A pole drops into the whisker field at specific locations for 1.3 s then retracts ('sample'). After another 1.3 s ('delay'), an auditory go cue is played, and the mouse licks the left or right lickspouts to report choice ('response'). The mouse is released after 60 s of head-fixation ('time-up release'). The mouse also self-releases by pressing against the floor ('self-release').
https://elifesciences.org/articles/66112#video1

a narrow spacing with shapes complementary to the headbar. Mice thus always entered the headport to reach the same head-restrained configuration. Upon entry, the headbar triggered two mechanical switches on both sides of the headport which activated the pneumatic pistons. At the end of the head-fixation, the pneumatic pistons were retracted, and the mouse was free to pull out from the headport. The release either came after a predefined duration for each head-fixation (up to 1 min, 'time-up release') or could be triggered by the mouse ('self-release') (*Video 1*).

Self-release was detected by a load-sensing platform (*Figures 2C* and *1C* insert). Continuous readings from a micro load cell reported weight on the platform and could be used to measure the mouse's daily body weight (*Figure 2C*) (adapted from *Noorshams et al., 2017*). During head-fixation, the weight on the platform decreased as a part of the weight was taken off by the headbar clamp (*Figure 2D*). The fluctuations in weight readings reflected mouse body movements. During struggles that typically indicated the mouse's efforts to get free from head-fixation, the weight readings produced either large negative or positive values that were far outside the normal range. A threshold was set to detect these struggle events and trigger self-release (*Figure 2D*). This threshold was adaptive: it gradually increased if struggle events were frequent or decreased if infrequent (Materials and methods).

We developed an operant conditioning algorithm to acclimate naive mice to voluntarily perform head-fixations in their home-cage (*Figure 2E*). Initially, the lickport was positioned close to the headport with the lickspouts inside the home-cage. Mice easily accessed the lickport and obtained water rewards upon licking. The rewarded lickspout alternated between the left and right lickspouts (three times each) to encourage licking on both. Gradually, the lickport retracted away from the home-cage (3 mm after every 20 rewarded licks) and mice were lured into the headport (*Figure 2F*). The lickport retraction stopped when mice entered deep into the headport to reliably trigger the head-fixation switches (Materials and methods). If no licks were detected for 12 hr, the program would re-extend the lickport closer to the home-cage to lure mice in again (*Figure 2F* top). During this phase of the training, the pneumatic pistons for head-fixation were not activated by the switches (*Figure 2E–F*, 'learn headport entry'). This was important to let mice first acclimate to the headport entry.

Once lickport retraction was completed, the pneumatic pistons were turned on (*Figure 2E–F*, 'learn head-fixation'). Head-fixation training started with soft clamp (low pistons pressure, 1.78 bar) and short duration (time-up release, 3 s). During head-fixation, mice could lick the lickspouts to obtain water reward. Gradually, the fixation duration was increased (2 s after every

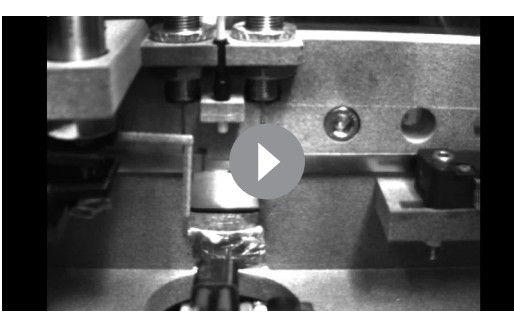

**Video 2.** Optogenetic photostimulation during task performance in home-cage. In a subset of trials, 630 nm light is turned on during either the sample or delay epoch. Photostimulation is through a clear skull implant to activate red-shifted opsins expressed in specific brain regions. During unsupervised optogenetic testing, the light source is positioned over the targeted brain region. In addition, a 630 nm masking flashing is given in every trial to prevent the mouse from distinguishing the trials with photostimulation. The masking flash is turned off in this example video for demonstration purposes.
https://elifesciences.org/articles/66112#video2

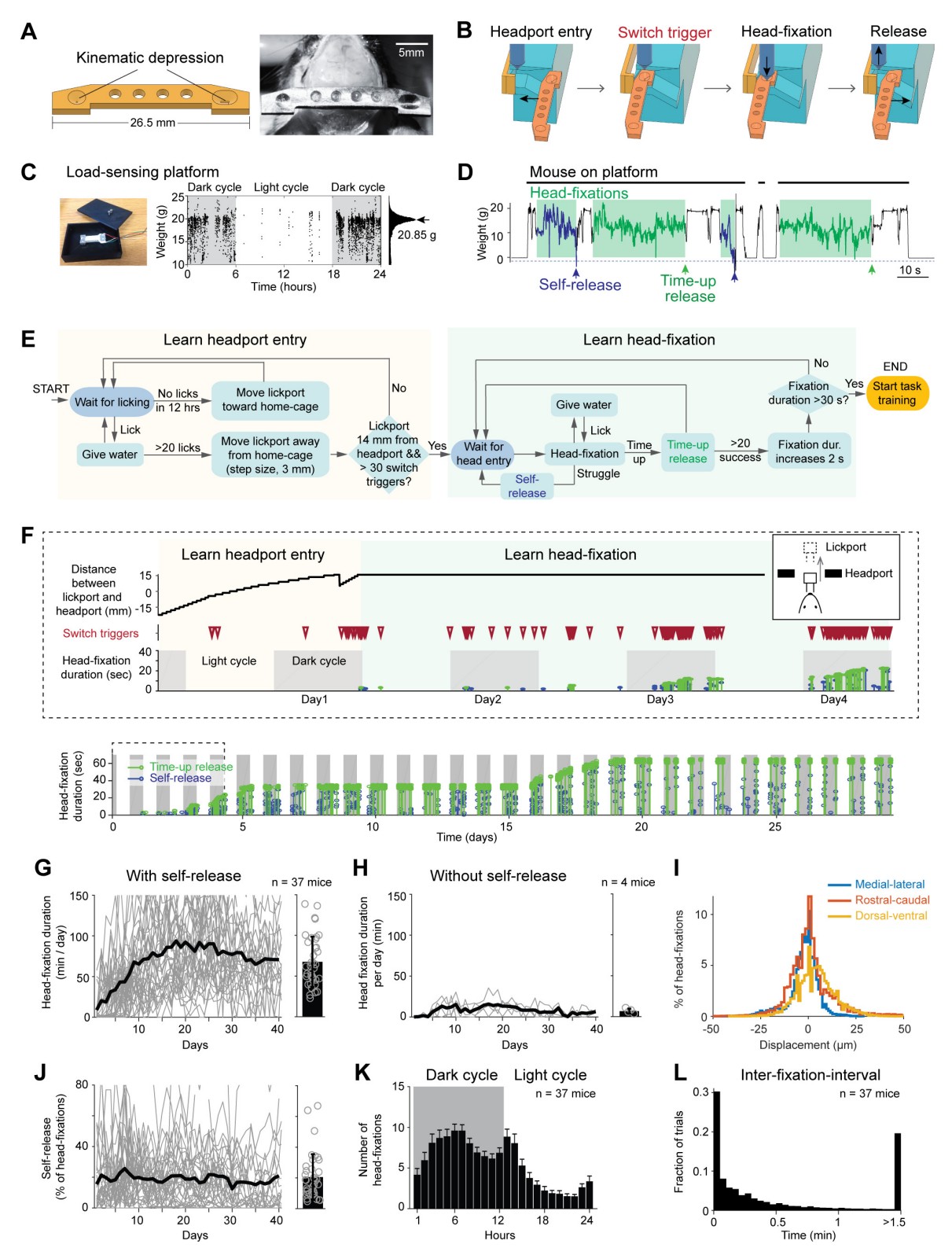

**Figure 2.** Voluntary head-fixation in home-cage. (**A**) Left, schematic drawing of the custom headbar. Right, photograph of a headbar implant. (**B**) Schematic drawings of a head-fixation and release sequence. Headbar enters a widened track on both sides of the headport that guides the headbar into a narrow spacing at the end. Two mechanical switches located on either side of the headport trigger pneumatic pistons to clamp the headbar. Head-fixations are released by retracting the pneumatic pistons. (**C**) Left, photograph of the load-sensing platform with top plate removed and load cell

*Figure 2 continued on next page*

*Figure 2 continued*

exposed. Right, example readings from the load cell (20 samples/s) in a 24-hr period. Shaded areas, dark cycles. Absence of samples indicates the mouse is off the platform. The histogram shows all readings from the 24-hr period. The peak can be used to estimate the mouse's body weight. (D) Example readings from the load cell during four consecutive head-fixations (green shades). Head-fixations typically reduce weight on the platform. Readings crossing a threshold (blue dashed line) result in self-release (blue arrows). Otherwise, the mouse is released after a predefined fixation duration (time-up release, green arrows). Fixation duration is 30 s in this example. (E) Flow chart of the head-fixation training protocol. See Materials and methods for details. (F) Data from an example mouse undergoing head-fixation training. Top, data from the first 4 days. The plots show lickport position (top, large value indicates further away from the home-cage, see inset), switch trigger events (middle), and head-fixation events (bottom). For head-fixation events, each tick indicates one fixation, with the height indicating fixation duration. The color indicates time-up release (green) and self-release (blue). Shaded areas, dark cycles. Time spent in learning headport entry and learning head-fixation are colored as in (E). Bottom: head-fixation data from the same mouse over 29 days. (G) Head-fixation duration over 40 days. Gray lines, individual mice; black line, mean. Bar plot shows average fixation duration throughout the entire head-fixation training. Error bar, standard deviation. Circles, individual mice. (H) Same as (G) but for mice without the self-release mechanism. (I) Displacement of the headbar implant across different head-fixations along medial-lateral, rostral-caudal, and dorsal-ventral directions. (J) Fraction of head-fixations in which mice trigger self-release. Gray line, individual mice; black line, mean. Bar plot shows average fraction throughout the entire head-fixation training. Error bar, standard deviation. Circles, individual mice. (K) Frequency of head-fixation across dark and light cycles. Bars show average across all mice. Error bars, standard deviations. (L) Time interval between head-fixations. Data from all mice are pooled.

The online version of this article includes the following figure supplement(s) for figure 2:

**Figure supplement 1.** Displacement of headbar implant across multiple head-fixations.

20 time-up releases). After the fixation duration reached 10 s, the pressure of the clamp also increased (hard clamp, 2.78 bar). Head-fixation training concluded after the fixation duration reached 30 s (*Figure 2E*). The fixation duration was further increased to 1 min at the late stage of task learning (*Figure 2F*, see task training below).

Under this protocol, mice quickly acclimated to the head-fixation (*Figure 2G*). Most mice (37/39) learned to self-engage in voluntary head-fixation and reached 30 s fixation duration in 7 ± 4.8 days (mean ± SD across mice). The total fixation duration per day increased monotonically over the first 10 days and plateaued at 69 ± 32.4 min per day (*Figure 2G*, 130 ± 56 fixations/day, mean ± SD). The self-release mechanism was critical for learning voluntary head-fixation. Without the self-release mechanism, the headport became aversive to mice after one unsuccess attempt to get free from head-fixation. Consequently, mice failed to learn voluntary head-fixation (*Figure 2H*). Highly trained mice continued to utilize self-release on 20.7 ± 14% of the head-fixations (*Figure 2J*). Most (67%) head-fixations occurred during the dark cycles (*Figure 2F and K*). Multiple head-fixations typically occurred in bouts, with majority of head-fixations occurring within a second apart (*Figure 2L*). The headbar position across multiple head-fixations was highly reliable (*Figure 2I* and *Figure 2—figure supplement 1*, |displacements| in medial-lateral, rostral-caudal, and ventral-dorsal dimensions, 6.4 ± 12, 8.8 ± 15 and 12.1 ± 14.7 μm, mean ± SD; Materials and methods).

Thus, mice can readily learn to perform repeated voluntary head-fixations for water reward. The extended duration of head-fixation makes behavioral task training possible.

## Autonomous training in a tactile decision task

We next integrated an algorithm to autonomously train mice in a tactile decision task with a short-term memory component (*Guo et al., 2014b*; *Guo et al., 2014a*; *Figure 3A*). During each head-fixation, mice were tested in a succession of trials. Each trial started with a sample epoch (1.3 s), in which mice were presented with a pole at one of two locations (anterior or posterior). The pole was always presented to the right whiskers. Mice were trained to discriminate pole location using their whiskers and report object location using directional licking (anterior location→ lick left, posterior location→ lick right). The sample epoch terminated when the pole moved out of reach, and mice were trained to withhold licking while remembering the choice during a delay epoch (1.3 s). At the end of the delay epoch, an auditory 'go' cue (100 ms) signaled the beginning of the response epoch and mice initiated licking to get water reward (*Figure 3B*). Incorrect responses led to a timeout. Premature licks before the 'go' cue were rare in trained mice and led to a brief timeout ('early lick', Materials and methods). Each trial was followed by an inter-trial-interval (2.5 s), after which the next trial began, until the head-fixation is released (*Video 1*).

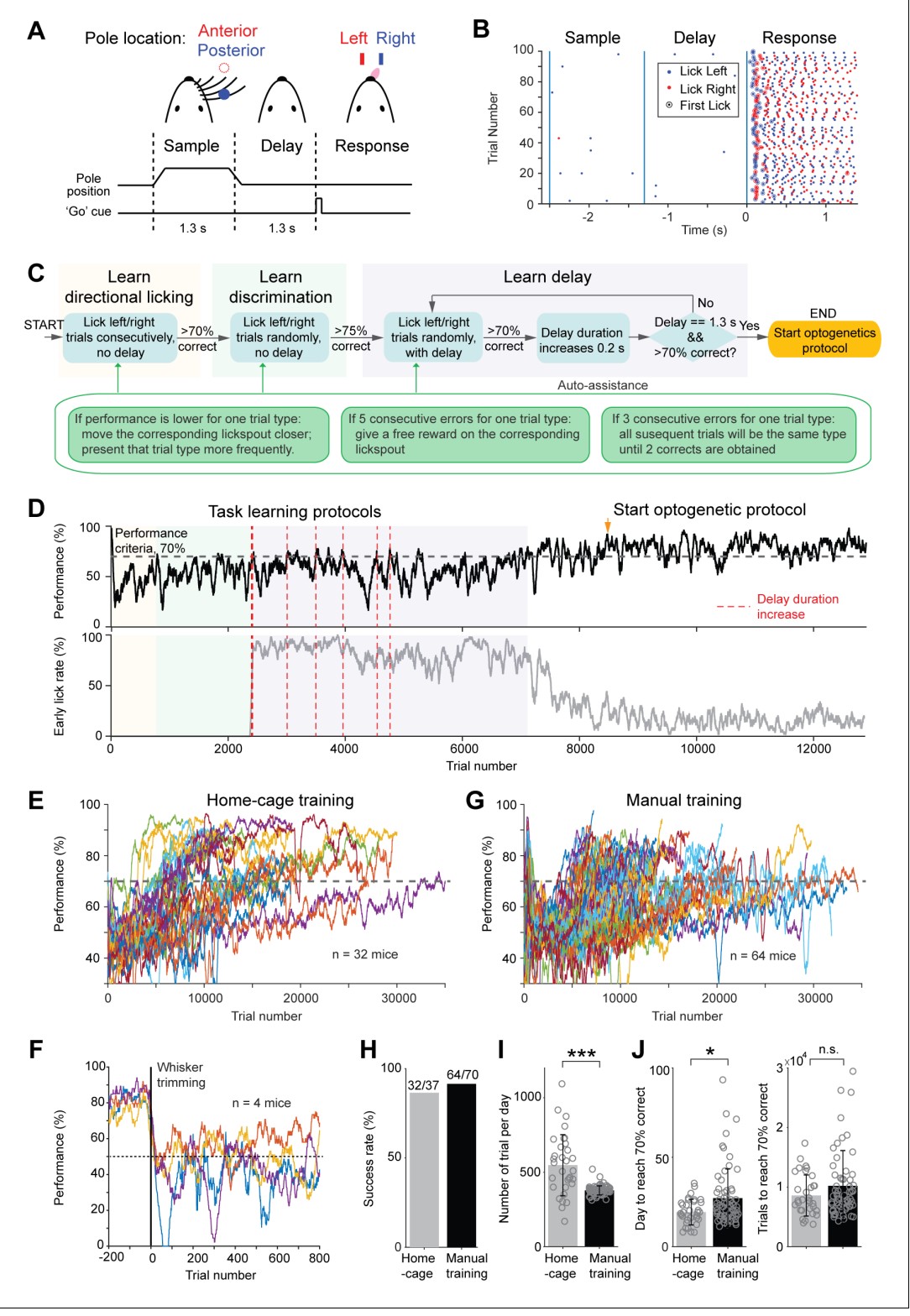

**Figure 3.** Tactile decision task in home-cage. (**A**) Task structure. Mice discriminate the location of a pole (anterior or posterior) during a sample epoch (1.3 s) and report the location using directional licking (left or right) after a delay epoch (1.3 s). An auditory go cue (0.1 s) signals the beginning of the response epoch. (**B**) Example behavioral data in 100 consecutive trials. Dots show individual licks. Blue, lick right; red, lick left. Circles indicate the first lick after the go cue (choice). In trials with early licks before the 'go' cue, choice licks occur late due to the

*Figure 3 continued on next page*

*Figure 3 continued*

timeouts (Materials and methods). (C) Flow chart of the task training protocol. See Materials and methods for details. Auto-assist programs (green box) evaluate mice performance continuously and assist mice whenever certain behavioral biases are detected. (D) Data from an example mouse undergoing task training in home-cage. Top, behavioral performance. Shaded areas indicate different phases of the training as in (C). During the delay epoch training, the red dash lines indicate delay duration increases. Bottom, fraction of trials in which the mouse licked before the go cue. After mice complete the task training protocol, experimenters examine the mice performance and initiate optogenetic testing protocol (indicated by the orange arrow in this example). (E) Behavioral performance of all mice in home-cage training (n = 32). Black dash line, criterion performance, 70% correct. (F) In a subset of mice (n = 4), the right whiskers were trimmed after home-cage training. Behavioral performance dropped to chance level (50%, black dash line) and did not recover. (G) Behavioral performance of all mice in manual training (n = 64). (H) Percentage of mice successfully trained in home-cage vs. manual training. Training is deemed successful if the mouse reached 70% correct criterion performance. (I) Number of trials performed per day in home-cage versus manual training. Bar plot shows mean and standard deviation across mice. Circles, individual mice. ***, p<0.001, two-tailed t-test. (J) Left, number of days to reach 70% correct criterion performance. Right, number of trials to reach 70% correct criterion performance. *, p<0.05; n.s., p>0.05, two-tailed t-test.

The online version of this article includes the following figure supplement(s) for figure 3:

**Figure supplement 1.** Behavioral performance of home-cage trained mice after transferring to an electrophysiology setup.

To facilitate learning, the automated algorithm divided task training into three phases (*Figure 3C*). The first phase started after mice learned to maintain head-fixation for 30 s (*Figure 2E*). In this phase (*Figure 3C*, 'learn directional licking'), lick left or lick right trials were presented consecutively and mice had to obtain three trials correct before the program switched trial type. This forced mice to lick both lickspouts. Once mice reliably switched lick direction across trial types, the program advanced to the second phase, in which the two trial types were presented randomly (*Figure 3C*, 'learn discrimination'). This required mice to discriminate object location to produce correct choice responses. During these early phases of training ('learn directional licking' and 'learn discrimination'), mice were free to lick at any time during the trial, but only the first lick after the 'go' cue were registered as choice (*Figure 3B*). When performance reached 75% correct, the final phase of the training enforced a delay epoch in which licking before the 'go' cue triggered a brief timeout (*Figure 3C*, 'learn delay', Materials and methods). The duration of the delay epoch was initially short (0.3 s), but it gradually increased to 1.3 s. Task training concluded when performance was stably above 70% correct. After task training concluded, the head-fixation duration was further increased from 30 s to 1 min before the start of optogenetic testing. This allowed more trials in each head-fixation.

We found that two factors were critical for successful home-cage training. First, mice must be acclimated to the task stimuli while learning voluntary head-fixation (*Figure 2E*), well before task training. During head-fixation training, the tactile stimulus and the auditory 'go' cue were presented upon each headport entry, even though the information was not required for successful performance (*Figure 2E*, Materials and methods). Second, mice often developed idiosyncratic biases by licking one lickspout more frequently, or sometimes continuously licking one lickspout without switching to the other. To counter these behavioral patterns, several 'auto-assist' programs were needed throughout task training (*Figure 3C*). The auto-assist programs evaluated mice performance and assisted the mice whenever certain behavioral patterns were detected (Materials and methods). Specifically, if a mouse licked one lickspout more frequently, the program moved the preferred lickspout further away from the mouse. When a mouse made consecutive errors for one trial type, the program presented that trial type more frequently or gave a free water reward on the correct lickspout. These measures countered biases and encouraged mice to switch lick direction across trial types.

Most mice (32/37, 87%) successfully learned the tactile decision task in automated home-cage training. *Figure 3D* shows the performance of an example mouse. Performance gradually improved during training. During introduction of the delay epoch, performance fluctuated as longer delays were progressively added (*Figure 3D*, red lines). Performance eventually increased and was stable over long periods of testing. Meanwhile, the number of early licks decreased. The learning speed was variable across individual mice (*Figure 3E*). Mice performed 547 ± 205 trials (mean ± SD) per

day in home-cage training and reached 70% correct in 19.3 ± 7.2 days (equivalent to 8588 ± 3453 trials). To confirm that mice solved the tactile decision task using their whiskers, we trimmed the whiskers in a subset of mice. Performance dropped to chance level after whisker trimming (*Figure 3F*). To examine whether home-cage training was robust to setup transfers, several mice were transferred to an electrophysiology setup after reaching criterion performance. Performance initially dropped, but it quickly recovered over 7 days (*Figure 3—figure supplement 1*). Thus, automated home-cage training could be used to support head-fixed electrophysiology or imaging experiments.

We compared the home-cage training to manual training supervised by experimenters. We trained a separate group of mice (n = 70) in daily sessions using conventional methods (*Guo et al., 2014b*). Mice were manually head-fixed and underwent daily training sessions (1–2 hr). The manual training followed a similar protocol as the home-cage training (Materials and methods). Learning speed and success rate were similar to the home-cage training (*Figure 3G–H*, 64/70 mice reached criterion performance vs. 32/37 in home-cage training; p=0.42, Chi-square test). Mice performed fewer number of trials per day in manual training (*Figure 3I*, 547 ± 205 vs. 377 ± 30 trials, automated vs. manual training, mean ± SD, p<0.001, two-tailed t-test). Consequently, manual training took more days to achieve performance criteria (*Figure 3J*, 19.3 ± 7.2 vs. 27.1 ± 16.3 days, p<0.05, two-tailed t-test), as mice took similar number of trials to reach criterion performance (*Figures 3J* and 8, 588 ± 3453 vs. 10,210 ± 5918 trials, p=0.39, two-tailed t-test).

These results show that mice could learn challenging perceptual decision tasks under head-fixation through unsupervised training in home-cage settings. Automated home-cage training has similar success rate and speed as manual training.

## A model-based comparison of task learning in automated and manual training

The home-cage system standardized the training across mice and continuously tracked mice behavior across the entire acquisition of the tactile decision task, thus providing an opportunity to examine task learning free of human interventions. We examined mice's behavioral strategies during task learning by modeling the choice behavior at various stages of training using logistic regression (Materials and methods). The model predicted mice's choice (lick left or lick right) from the tactile stimulus, stimulus history, choice history, reward history, a win-stay-lose-switch strategy (choice x reward in the previous trial), and a constant bias (*Figure 4A*).

The model was able to predict mice's behavioral choice across different stages of training (*Figure 4B–C*). Interestingly, the model could predict choice well before the behavioral performance was above chance (*Figure 4B–C*). This suggests that mice used behavioral strategies other than the tactile stimulus to guide choice during the early phase of training. To determine which model regressor was driving choice, we built partial models that excluded individual regressors and compared their prediction accuracy to the full model (*Figure 4B,p* value indicates significantly worse prediction than the full model based on cross validated performance, bootstrap, Materials and methods). Model selection showed that two regressors most strongly contributed to choice prediction, but these regressors contributed at different stages of training (*Figure 4B–C*). During the early phase of training, choice history from the last trial had a significant contribution, suggesting that mice tended to repeat their choice regardless of the tactile stimulus. During the late phase of training, the contribution of choice history diminished, and the contribution of the tactile stimulus increased, which suggests that mice learned to use the tactile stimulus to guide choice (*Figure 4B* bottom, 4C). A model that only considered choice history and tactile stimulus was sufficient to account the choice prediction performance of the full model (*Figure 4C*).

This pattern of behavioral strategy was consistently observed in home-cage training (*Figure 4D* top). A similar pattern of behavioral strategy was also observed in manual training (*Figure 4D* bottom). Overall, similar percentages of mice in home-cage and manual training used the tactile stimulus, choice history, and reward history to solve the task during learning (*Figure 4E*). These results suggest that naive mice initially adapted a behavioral strategy of repeating their past actions, and then abandoned this strategy as they learned the sensorimotor contingency. These results show that mice in home-cage training used similar behavioral strategies to learn the tactile decision task as mice in manual training. This provides further validation data that shows automated training can replace conventional manual training.

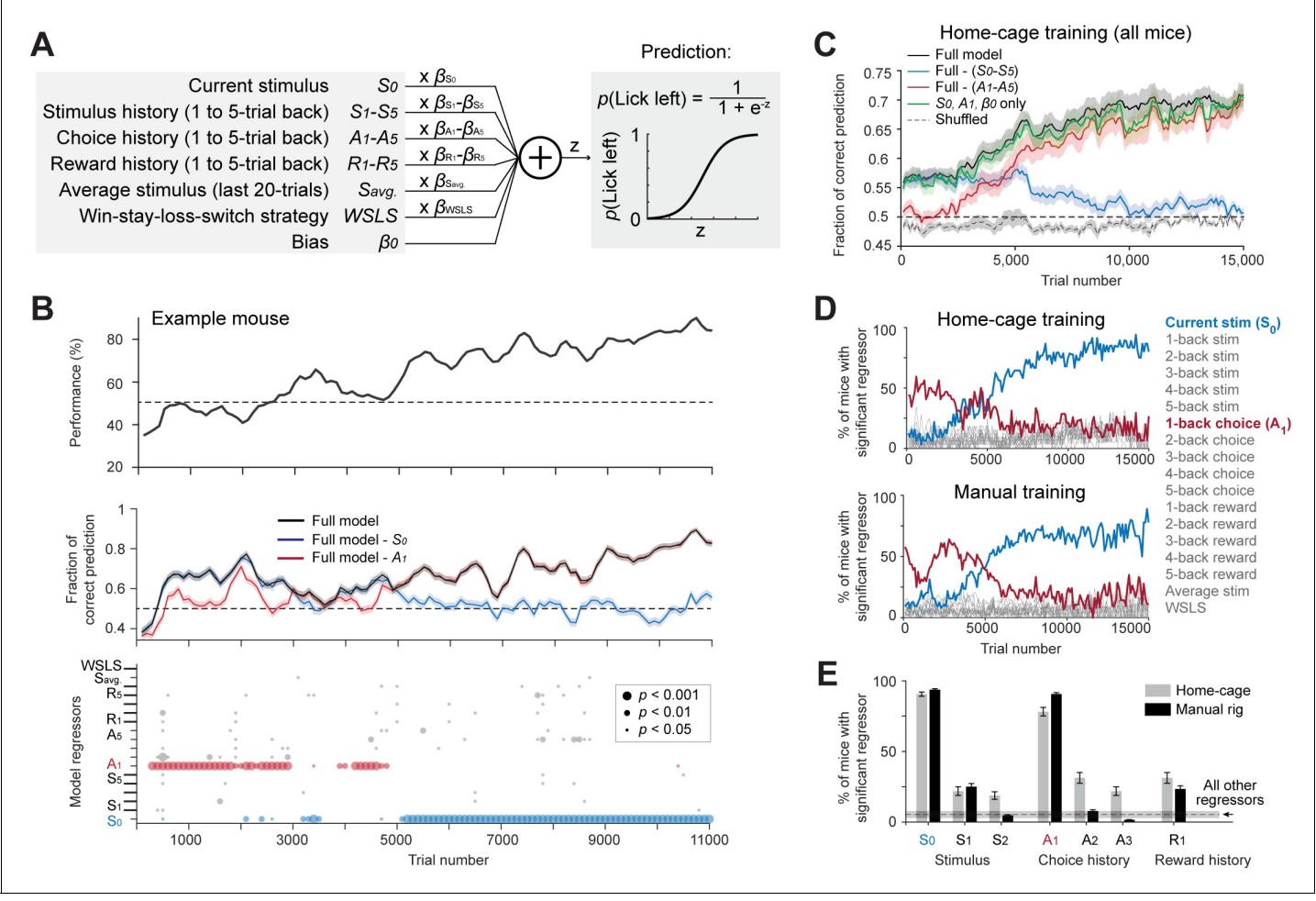

**Figure 4.** Model-based comparison of task learning in home-cage and manual training. (**A**) A logistic regression model to predict choice. Weighted sum of the tactile stimulus, stimulus history, choice history, reward history, a win-stay-lose-switch strategy (choice x reward in the last trial), and a constant bias is passed through a logistic function to predict choice in the current trial. (**B**) Behavioral data and model prediction from an example mouse in home-cage training. Trials are binned (bin size, 500 trials; step size, 100 trials). Top, behavioral performance. Middle, prediction performance of the full model and two partial models excluding either the current stimulus ($S_0$, blue) or 1-back choice ($A_1$, red). Model performance is calculated as the fraction of choice predicted (Materials and methods; chance level is 50%). Shaded area indicates SEM. Bottom, the significance of individual regressors. Circle size corresponds to p values. The significance of a regressor is evaluated by comparing the prediction of the full model to a partial model with the regressor of interest excluded. p Values are based on bootstrap (Materials and methods). (**C**) Average model prediction across all mice in home-cage training. Black line, prediction of the full model. Blue, performance of a partial model excluding both the current stimulus $S_0$ and stimulus history $S_{1-5}$. Red, performance of a partial model excluding choice history $A_{1-5}$. Green, performance of a partial model with only the current stimulus $S_0$, choice history $A_1$, and a constant bias term $\beta_0$. Dashed line, the performance of the full model predicting shuffled behavioral data (Materials and methods). Shaded area indicates SEM across mice. Chance, 50%. (**D**) Percentage of mice showing significant contribution from each regressor at different stages of learning. Significance is defined as p<0.05. Top, mice in home-cage training ($n = 32$); Bottom, mice in manual training ($n = 64$). (**E**) Percentage of mice relying on different regressors during task learning. A mouse is deemed to rely on a regressor if it shows significant contribution to choice prediction in at least five consecutive time bins during training (1000 trials). Regressors shown are the current stimulus ($S_0$), 1-back and 2-back stimulus history ($S_{1-2}$), 1-back, 2-back and 3-back choice history ($A_{1-3}$), and 1-back reward history ($R_1$). Error bars show SEM across mice (bootstrap). Dash line and shaded area show the mean and SEM across all other regressors. All other regressors show small contributions and they are pooled. Regressors from both home-cage and manual training are also pooled.

## Contingency reversal learning

The automated home-cage system permits prolonged task training, which opened the possibility of training mice in challenging behavioral tasks that were previously difficult to attain. To test this utility, we trained mice in contingency reversals in which they had to flexibly report the tactile decision using lick left or lick right (*Figure 5A*).

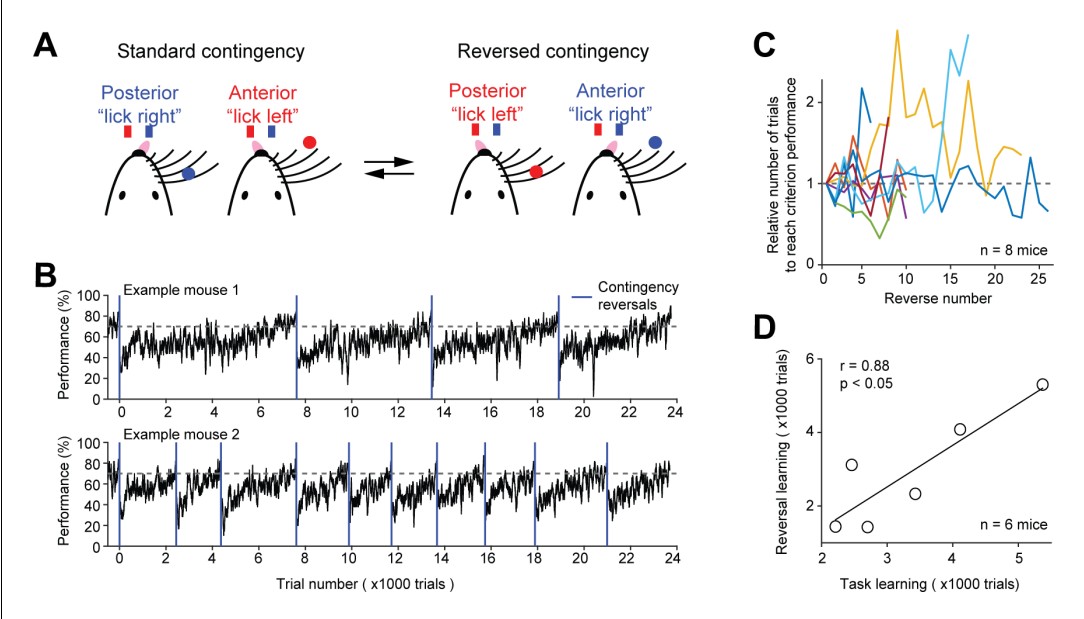

**Figure 5.** Contingency reversal learning in home-cage. (**A**) Mice discriminate the location of a pole (anterior or posterior) and report the location using directional licking (left or right) without a delay epoch. The task switches between standard sensorimotor contingency and reversed contingency once mice reach criterion performance. Criterion performance, >80% for 100 trials. (**B**) Behavioral performance data from two example mice. Bin size, 50 trials. Blue line, contingency reversals. Dashed line, 70% correct. (**C**) The number of trials to acquire new contingencies over multiple contingency reversals. The number of trials to reach criterion performance is normalized to the first contingency reversal. Individual lines show individual mice. (**D**) The number of trials needed to learn the tactile decision task vs. the average number of trials to reach criterion performance in contingency reversal learning. Task learning is from the start of head-fixation training to reaching criterion performance. Individual dots show individual mice. Line, linear regression. Two mice from (**C**) were excluded because they previously learned a different behavioral task.

Mice first learned the standard tactile decision task (without a delay) in which they reported anterior pole location by licking left and posterior pole location by licking right. After mice attained high levels of performance (>80% correct for 100 trials), the sensorimotor contingency was reversed in which anterior pole location corresponded to lick right and posterior pole location corresponded to lick left. Mice did not receive any cues about the reversal other than reward feedbacks: correct responses led to water rewards; incorrect responses led to timeouts. Immediately after the reversal, behavioral performance dropped to below chance (*Figure 5B*). Performance steadily recovered and was eventually stably above 70% correct.

To examine whether mice could robustly switch sensorimotor contingency, we repeatedly reversed the contingency after mice reached criterion performance. Mice consistently acquired new contingencies and did so in similar number of trials (*Figure 5B–C*). However, the reversal learning speed varied substantially across mice (*Figure 5B*). The initial task acquisition speed (i.e. the number of trials to reach criterion performance from the start of head-fixation training) was correlated with the reversal learning speed (i.e. the number of trials to reach criterion performance after contingency reversal) (*Figure 5D*). Thus, mice could be screened for fast learners based on the initial task acquisition speed.

These data, together with the robust training in the tactile decision task with short-term memory (*Figure 3*), demonstrate the utility of prolonged home-cage training in teaching mice difficult decision-making tasks.

## Home-cage testing reveals behavioral signatures of motivation

In home-cage experiments, mice behavior was motivated by water rewards. Mice received all their daily water by engaging in the task. We examined mice's water consumption and body weight during home-cage training. When water restricted mice were introduced into the home-cage system, all mice obtained a large number of rewards on day 1 (*Figures 6A–B*, 600 rewards on average, 1.8 mL of water). This was likely due to the ease of accessing the lickport (see Voluntary head-fixation in

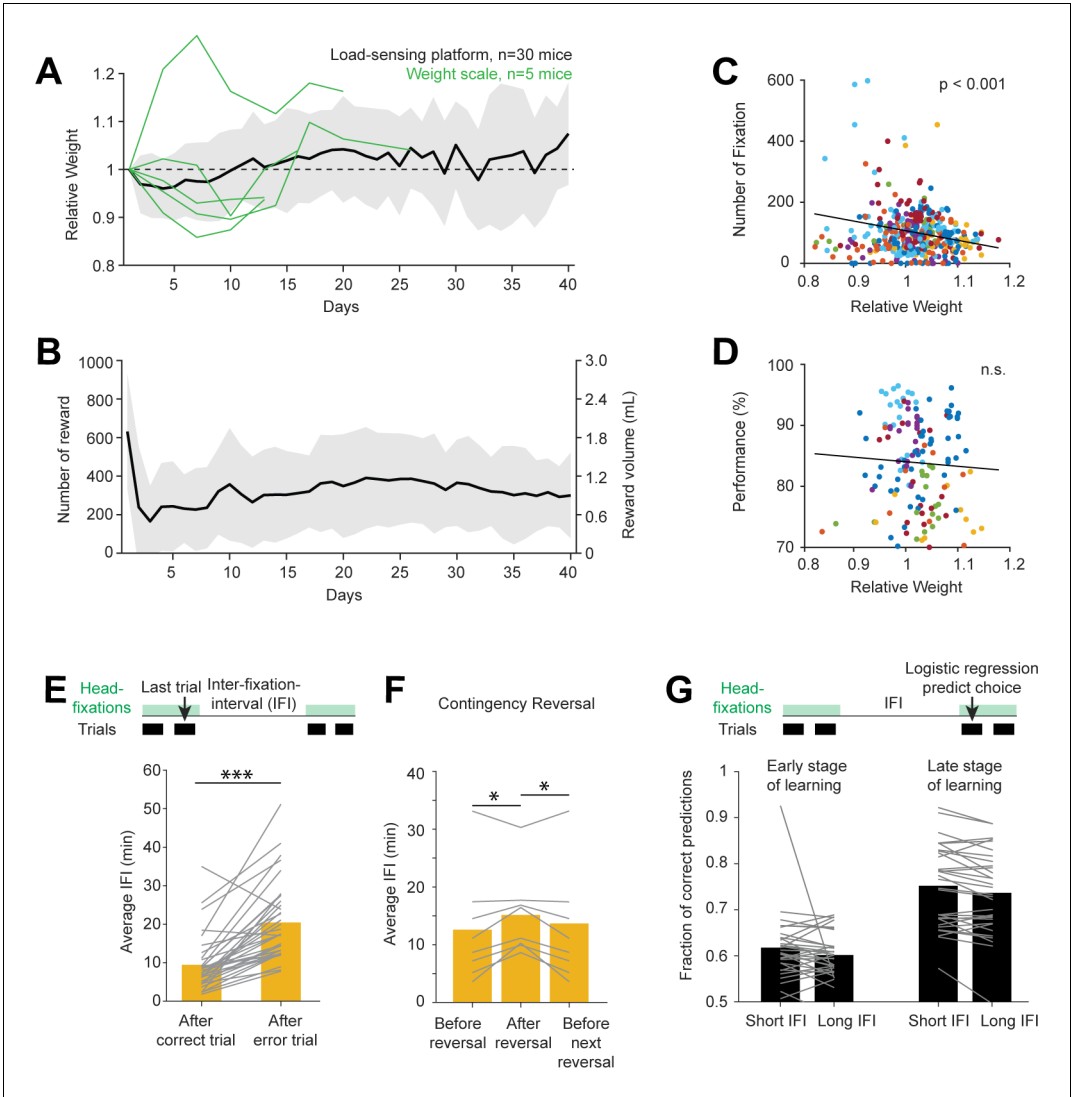

**Figure 6.** Contingency reversal learning in home-cage. (**A**) Mouse weight as a function of time. Body weights were estimated from the load-sensing platform in home-cage (see *Figure 2C*). Weights are normalized to the initial weights on day 1. Black line and shades, mean ± standard deviation across mice. Green lines, in a subset of mice, body weights were also measured outside of the home-cage on a weight scale. (**B**) Number of rewards and water consumed per day. Line and shades, mean ± standard deviation across mice. (**C**) Number of head-fixations per day as a function of normalized body weights. Each symbol corresponds to one day. Different colors show different mice. (**D**) Task performance as a function of normalized body weights. Multiple factors can affect task performance, including motivation and task learning. Here, the data are taken from days after the mice have reached criterion performance. (**E**) Average IFI durations following correct and error trials. Individual lines show individual mice. Bars show averages across mice. ***p<0.001, paired two-tailed t-test. (**F**) Average IFI durations during contingency reversal learning. Trials are taken from periods right before contingency reversals, immediately following reversals, and before the next reversals. Individual lines show individual mice. Bars show averages across mice. *p<0.05, paired two-tailed t-test. (**G**) Prediction of choice by logistic regression on trials following long vs. short inter-fixation-intervals. The logistic regression model was fit using trials in their natural sequential order (regardless of the inter-fixation-intervals). The model was then used to predict choice on independent trials. Trials were then sorted by the preceding inter-fixation-intervals. Prediction performance was calculated separately for trials following short or long inter-fixation-intervals. Individual lines show individual mice. Bars show averages across mice. n.s. p>0.05, paired two-tailed t-test. (**H**).

home-cage, *Figure 2E*). As the lickport was retracted into the headport (away from the home-cage), reward rate dropped significantly on subsequent days. Water consumption and body weight gradually increased after the initial dip as mice acclimated to the head-fixation (*Figure 6A–B*). At steady state, a mouse typically consumed ~1 mL of water daily in the home-cage while maintaining stable body weight. This amount of water consumption was similar to mice engaged in daily manual experiments (*Guo et al., 2014b*). The number of head-fixations per day was correlated with body weight (*Figure 6C*). Since body weight reflected prior water consumption, this indicates different levels of motivation due to thirst, which drove engagement in the task. In highly trained mice, task performance was stable despite the body weight change (*Figure 6D*).

We inferred mice's motivation to engage in the task by examining the time intervals between head-fixations ('inter-fixation-interval'). We sorted the inter-fixation-intervals by the outcome of the last trial in the previous head-fixation. The inter-fixation-interval after an error (which led to no reward) was significantly longer than following a correct trial (*Figure 6E*). This indicates a loss of motivation after an error, perhaps due to the loss of an expected reward. Consistent with this interpretation, we also found a significant increase in inter-fixation-intervals shortly after a sensorimotor contingency reversal (*Figure 6F*). This coincided with a drop in task performance due to the rule change (*Figure 5B*). As performance recovered, inter-fixation-intervals also decreased (*Figure 6F*).

Despite the motivational change, mice maintained the same strategy in their choice behavior. To examine this, we used the logistic regression model to predict choice on trials following short vs. long inter-fixation-intervals (*Figure 6G*). If behavioral strategy changed across motivational state (reflected in short vs. long inter-fixation-intervals), the predictive power of the model would differ between these conditions. However, we did not find a significant difference in the model prediction performance. The result was similar in early and late stages of task learning (*Figure 6G*), even though mice used distinct strategies during these periods (*Figure 4*). These results suggest consistent strategies in the choice behavior.

Together, these results show behavioral signatures of motivation in self-initiated behavior in home-cage, which could be potentially exploited in studies of goal-directed behavior.

## Unsupervised home-cage optogenetic experiment

We integrated optical components into the behavioral test chamber (*Figure 1C*) to perform optogenetic manipulations during home-cage behavior. We used red light (630 nm) to photostimulate targeted brain regions through a clear skull implant (*Figure 7A*, Materials and methods) (*Guo et al., 2014a*). Red light is less subject to hemoglobin absorption (*Svoboda and Block, 1994*; *Tromberg et al., 2000*) and can penetrate neural tissues in vivo with less attenuation compared to blue or green light while producing less heating (*Liu et al., 2015*; *Stujenske et al., 2015*; *Wiegert et al., 2017*). A light source was mounted above the headport to broadly illuminate the targeted brain region (*Figure 1C*). To manipulate activity specifically in the targeted brain regions, we locally expressed red-shifted opsins, ChrimsonR (*Klapoetke et al., 2014*), or ChRmine (*Marshel et al., 2019*). This approach did not require optical fiber implants. Thus, it eliminated the need to manually couple a light source to the mouse and enabled continuous optogenetic testing without human interventions. Importantly, head-fixation provided stable access to the brain for repeatable optical stimulations.

We first tested this optogenetic strategy in the barrel cortex (vS1) for a well-documented channelrhodopsin-assisted photoinhibition method (*Cardin et al., 2009*; *Olsen et al., 2012*; *Li et al., 2019*). We injected small volumes (200 nL) of cre-dependent AAV viruses carrying either ChrimsonR or ChRmine in GAD2-IRES-cre mice (*Taniguchi et al., 2011*) to excite GABAergic neurons and inhibit nearby pyramidal neurons. Virus injection localized the opsin expression (*Figure 7A*, diameter of expression, 0.79–1.18 mm). We characterized this photoinhibition using silicon probe recordings in awake non-behaving mice under the same illumination conditions as in the home-cage (*Figure 7B*). Units with narrow spikes were putative fast spiking (FS) neurons (*Cardin et al., 2009*; *Olsen et al., 2012*; *Guo et al., 2014a*; *Resulaj et al., 2018*; *Li et al., 2019*) and a subset of the FS neurons were activated by light (*Figure 7C–D*, 7/14 with significantly elevated spike rate at 2.8 mW/mm$^2$, p<0.01, two-tailed t-test, photostimulation vs. baseline epoch). Neurons with wide spikes were likely mostly pyramidal neurons and majority of these neurons were silenced in a dose-dependent manner (*Figure 7C–D*, 114/157 with significantly depressed spike rate at 2.8 mW/mm$^2$). Photoinhibition

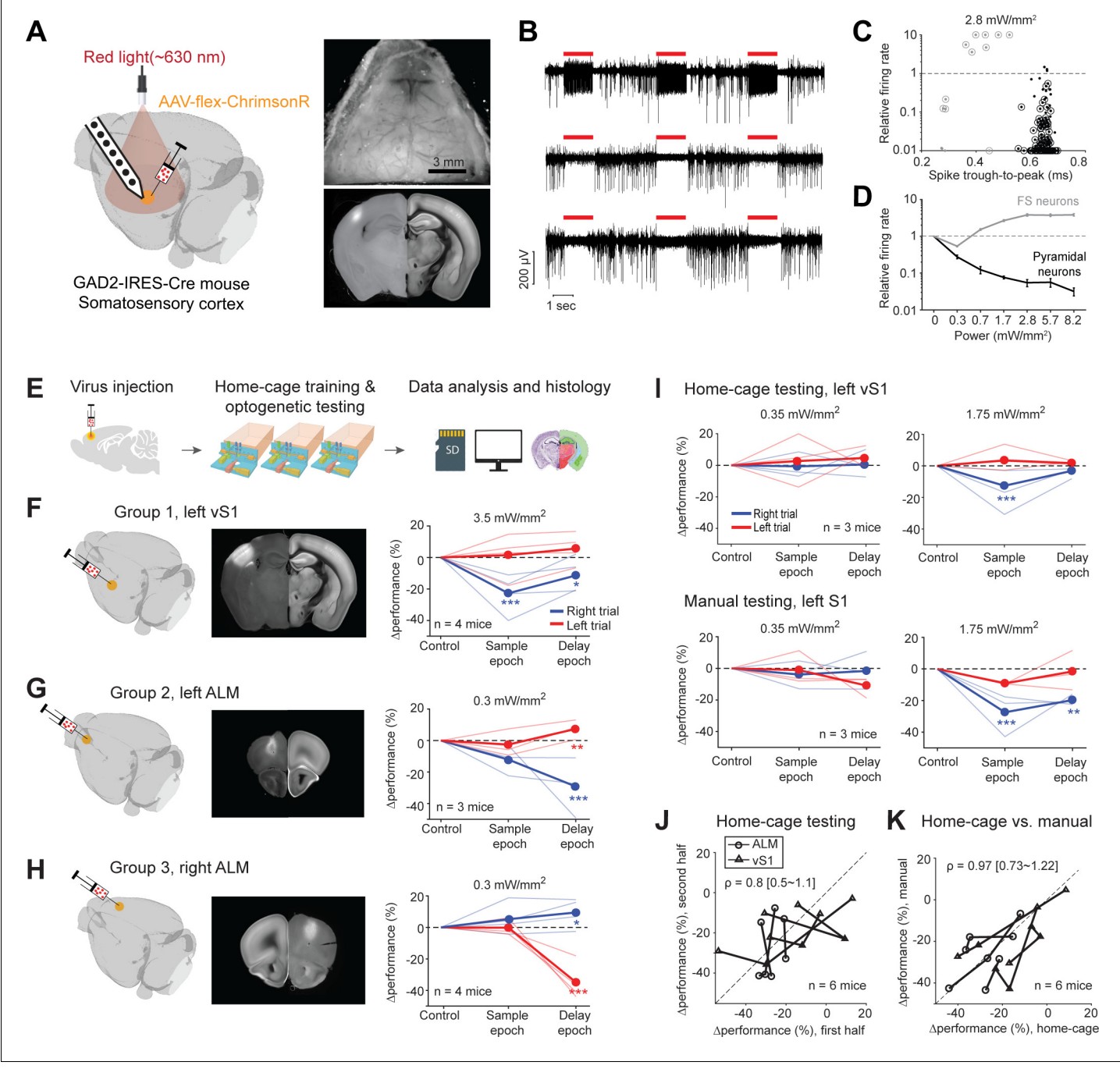

**Figure 7.** Photoinhibition of cortical regions and comparisons of home-cage optogenetic experiments with manual optogenetic experiments. (**A**) Left, an optogenetic approach to silence activity in specific brain regions and electrophysiology characterization in the barrel cortex (vS1). Right top, an example clear skull implant. Right bottom, a coronal section showing ChrimsonR expression in vS1. The coronal section is aligned to the Allen Refence Brain (Materials and methods). (**B**) Silicon probe recording in vS1 during photostimulation. Multi-unit activity from three example channels showing photoexcitation (first row) and photoinhibition (second and third rows). Red lines, photostimulation. (**C**) Effects of photostimulation on cell types defined by spike waveform. Dots, individual neurons. Circled dots, neurons with significant spike rate change, p<0.05, two-tailed t-test. Spike rate of each neuron during photostimulation is normalized to its baseline ('relative firing rate', Materials and methods). Neurons with narrow spike waveforms are putative fast-spiking (FS) interneurons (gray). Neurons with wide spike waveforms are putative pyramidal neurons (black). (**D**) Relative firing rate of putative pyramidal neurons (black) and interneurons (gray) as a function of photostimulation intensity. Error bars show SEM across neurons. (**E**) Workflow schematics. (**F**) Photoinhibition of the left vS1. Left, a 3D rendered brain showing virus injection location. Middle, a coronal section showing virus expression in the left vS1. Right, behavioral performance change relative to the control trials during photoinhibition in the sample and delay epoch. Performance for lick left (red) and lick right trials (blue) are computed separately. Thin lines, individual mice; thick lines, mean. *p<0.025;

*Figure 7 continued on next page*

*Figure 7 continued*

\*\*p<0.01; \*\*\*p<0.001, significant performance change compared to the control trials (bootstrap, Materials and methods). (**G**) Same as (**F**) but for photoinhibition of the left ALM. (**H**) Same as (**F**) but for photoinhibition of the right ALM. (**I**) Behavioral performance change relative to the control trials during photoinhibition in home-cage optogenetic experiments (top row) and manual optogenetic experiments (bottom row). See the full dose response in *Figure 7—figure supplement 2*. (**J**) Comparison of performance change during the first vs. second half of optogenetic testing. Data from all mice and experiments (left vS1 photoinhibition, three mice; left ALM photoinhibition, one mouse; right ALM photoinhibition, two mice). Lines connect data from multiple photostimulation intensities for individual mice. For each brain region, only the condition in which photoinhibition induced the largest behavioral effect is included. Left vS1, data from the lick right trials, sample epoch photoinhibition. Left ALM, data from the lick right trials, delay epoch photoinhibition. Right ALM, data from the lick left trials, delay epoch photoinhibition. Linear regression, slope: 0.8; range: 0.5–1.1 (95% confidential interval). There is no difference between the first and second half of the home-cage optogenetic experiments (p=0.78, paired t-test). Home-cage optogenetic experiments span 12 ± 4.5 days, mean ± SD. (**K**) Comparison of performance change in home-cage versus manual optogenetic experiments. Linear regression, slope: 0.97; range: 0.73–1.22 (95% confidential interval). There is no difference between home-cage and manual experiments (p=0.36, paired t-test).

The online version of this article includes the following figure supplement(s) for figure 7:

**Figure supplement 1.** Comparisons of effect size with previous studies.

**Figure supplement 2.** Behavioral performance change dose-response curve during optogenetics.

silenced >70% of the spikes in putative pyramidal neurons at the virus injection site over a wide range of laser powers (*Figure 7D*, 0.3–8.2 mW/mm$^2$).

We next tested the feasibility of unsupervised home-cage optogenetic experiments. Cortical regions involved in decision-making can vary across behavioral strategies and training conditions (*Chowdhury and DeAngelis, 2008*; *Liu and Pack, 2017*; *Gilad et al., 2018*). We examined whether behaviors resulting from automated home-cage training engaged the same cortical regions as manual training. We photoinhibited activity in two cortical regions known to be involved in tactile decision-making. We targeted the left vS1, contralateral to the side of the tactile stimulus, where photoinhibition was expected to impair tactile sensation (*O'Connor et al., 2013*; *Sachidhanandam et al., 2013*; *Guo et al., 2014a*). In addition, we targeted anterior lateral motor cortex (ALM), where unilateral photoinhibition was expected to bias choice to the ipsilateral direction (*Guo et al., 2014a*; *Li et al., 2015*). After mice reached high levels of performance in home-cage training, photostimulation was deployed in a subset of trials during either the sample or delay epoch (*Figure 7E*).

Photoinhibition of the left vS1 reduced behavioral performance primarily during the sample epoch (*Figure 7F*). The performance deficit was limited to lick right trials, which corresponded to the posterior pole position where the pole strongly contacted the whiskers. This pattern of behavioral effect is consistent with a deficit in pole detection (*O'Connor et al., 2013*; *Sachidhanandam et al., 2013*; *Guo et al., 2014a*). Photoinhibition of ALM produced an ipsilateral bias, primarily during the delay epoch (*Figure 7G–H*). Photoinhibition of the left ALM biased upcoming licking to the left, resulting in lower performance in lick right trials and slightly higher performance in lick left trials. An opposite bias was induced by photoinhibiting the right ALM. These patterns of behavioral deficit were similar to those observed in previous studies (*Guo et al., 2014a*; *Li et al., 2015*) and the effect size was comparable (*Figure 7—figure supplement 1*). As a negative control, photostimulation produced no effect when only GFP viruses were injected into ALM (*Figure 7—figure supplement 2A*).

Home-cage optogenetic experiments lasted 12 days on average (SD, 4.5 days). Mice showed little adaptation to photostimulation. Later days of the home-cage optogenetic experiments elicited similar effect sizes as the early days (*Figure 7J*). To directly compare the behavioral effects from home-cage testing to those induced in manual experiments, we subsequently tested a subset of mice (n = 6) in conventional optogenetic experiments. In daily supervised sessions, the mice were manually head-fixed and tested for photoinhibition on a different setup (Materials and methods). vS1 photoinhibition in manual experiments elicited the same pattern of behavioral deficit as those induced in home-cage testing (*Figure 7I and K*). The magnitude of behavior performance deficit was similar across a wide range of light doses (*Figure 7—figure supplement 2B*). Similar results were also obtained for ALM photoinhibition (*Figure 7K*).

These characterization data show that the optogenetic approach can potently manipulate cortical activity and unsupervised home-cage optogenetic experiments can be used to screen for cortical regions involved in behavior.

## Survey of subcortical regions involved in tactile decision-making

We next tested the optogenetic strategy in manipulating activity of deep brain regions. We focused on the action-selection networks that include the striatum and downstream superior colliculus (SC). Previous studies in rodents suggest both the striatum and SC play roles in perceptual decision-making based on olfactory, auditory, or visual cues (*Felsen and Mainen, 2008*; *Felsen and Mainen, 2012*; *Stubblefield et al., 2013*; *Znamenskiy and Zador, 2013*; *Duan et al., 2015*; *Kopec et al., 2015*; *Sippy et al., 2015*; *Yartsev et al., 2018*; *Duan et al., 2019*; *Lee et al., 2020*). However, the previous studies examined different subregions of the striatum and SC in different perceptual decision behaviors, making comparisons across studies difficult. We therefore compared striatal and SC subregions' involvement in the tactile decision behavior.

We injected cre-dependent ChRmine viruses into the left striatum of GAD2-IRES-cre mice to perturb GABAergic neurons non-specifically in the targeted region. We first tested if photostimulation through an intact clear skull could manipulate activity deep in the brain. We performed silicon probe recordings around an injection site 2.2 mm below the brain surface (*Figure 8A–B*). Most striatal neurons near the injection site were significantly excited or inhibited by photostimulation through the clear skull (*Figure 8C*). The mixture of excitation and inhibition was expected since the ChRmine viruses targeted GABAergic neurons non-specifically (*Taniguchi et al., 2011*), and the GABAergic striatal projection neurons and interneurons locally inhibit each other (*Burke et al., 2017*). For neurons modulated by light (p<0.01, two-tailed t-test), the changes in spike rate monotonically increased as a function of laser power (*Figure 8D*). Significant spike rate changes were observed even at low laser powers (3 mW or 1.75 mW/mm$^2$ on the brain surface). The effect was spatially localized to the injection site (*Figure 8E*). These data show that the optogenetic method can potently manipulate striatal activity.

We next tested if the striatal optogenetic manipulation was sufficient to bias behavior. We targeted three subregions of the striatum previously implicated in different types of decision-making behaviors, including a subregion of the anterior dorsal striatum (*Yartsev et al., 2018*), a subregion of the dorsolateral striatum (*Sippy et al., 2015*), and a subregion of the posterior dorsal striatum (*Znamenskiy and Zador, 2013*; *Figure 8F–H* and *Figure 8—figure supplement 1A–C*). Among them, the dorsolateral striatal subregion received inputs from both ALM (*Hooks et al., 2018*) and the barrel cortex (*Sippy et al., 2015*). The anterior striatum received inputs from only ALM and the posterior dorsal striatum did not receive inputs from either cortical regions (*Hooks et al., 2018*). We targeted the left striatum, ipsilateral to the left barrel cortex and contralateral to the tactile stimulus. Moreover, we targeted the striatal regions unilaterally to examine their roles in directional licking. Perturbation of the three striatal subregions differentially affected task performance (*Figure 8F–H* and *Figure 7—figure supplement 2C–E*). The performance deficits induced by perturbing the anterior and posterior striatum were minimal and limited to the delay epoch (*Figure 8F and H*; *Figure 7—figure supplement 2C and E*). In contrast, perturbing the dorsolateral striatum produced large performance deficit in both the sample and delay epochs, but not the response epoch (*Figure 8G* and *Figure 7—figure supplement 2D*). These patterns of behavioral deficit suggest that the dorsolateral stratum was required for tactile-guided licking decisions (*Sippy et al., 2015*).

Additionally, we examined SC downstream of the basal ganglia. We targeted a lateral region of SC previously implicated in the control of licking movement (*Rossi et al., 2016*; *Lee et al., 2020*; *Figure 8I* and *Figure 8—figure supplement 1D*). Activity in the lateral SC is thought to drive contralateral licking (*Lee et al., 2020*). We injected cre-dependent ChrimsonR (or ChRmine) viruses into the left SC in GAD2-IRES-cre mice and activated SC GABAergic neurons to photoinhibit SC output (*Duan et al., 2019*). Silicon probe recordings show that photostimulation modulated activity in the targeted SC region even at moderate laser powers (*Figure 8J–K*). SC neurons activated by light were presumably GABAergic neurons and they inhibited other SC neurons (*Figure 8L*). Silencing the left SC biased upcoming licking to the left, resulting in performance decrease specifically in lick right trials (*Figure 8M*). The effect was elicited by photoinhibition during the delay epoch, but not during the sample epoch (*Figure 8M*). The bias was light-dose dependent and was significant at moderate laser power (*Figure 7—figure supplement 2F*). These behavioral effects qualitatively mirrored those

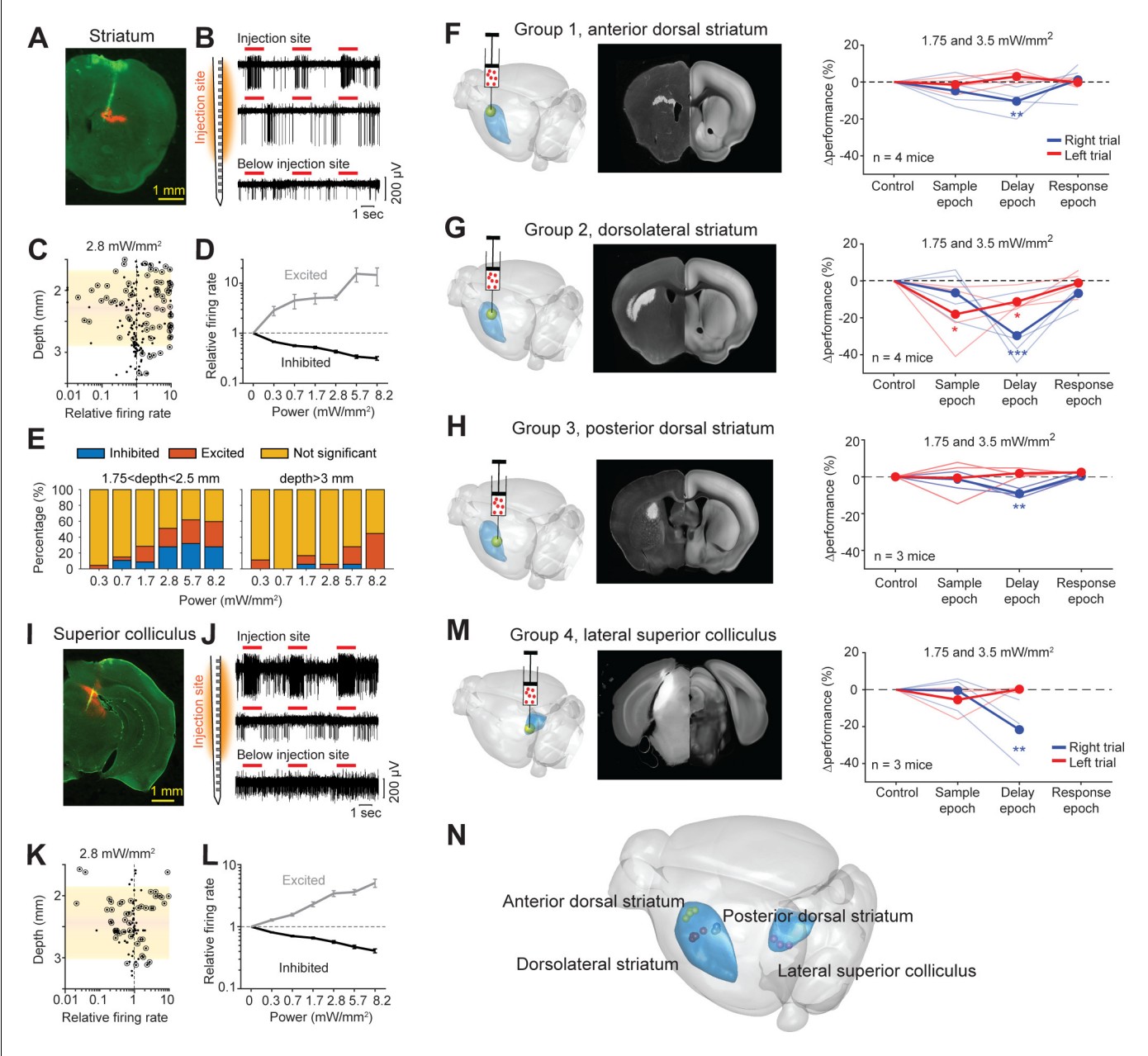

**Figure 8.** Photostimulation of subcortical regions in home-cage optogenetic experiment. (**A**) A coronal section showing virus expression in the striatum (red) and silicon probe recording track (green). (**B**) Silicon probe recording in the striatum during photostimulation. Multi-unit activity from two example channels near the virus injection site (top) and one example channel below the injection site. Red lines, photostimulation. (**C**) Effects of photostimulation across depths. Dots correspond to individual neurons. Circled dots indicate neurons with significant spike rate change, p<0.05, two-tailed t-test. Spike rate of each neuron during photostimulation is normalized to its baseline ('relative firing rate', Materials and methods). Shaded area indicates the virus expression region estimated from histology. (**D**) Relative firing rate of all significantly excited and inhibited neurons as a function of photostimulation intensity. Error bars show SEM across neurons. (**E**) Fraction of neurons significantly excited (red) and inhibited (blue) by photostimulation, p<0.05, two-tailed t-test. Left, neurons from near the virus injection site. Right, neurons from below the virus injection site. (**F**) Left, a 3D rendered brain showing the striatum (blue) and the injection location in the anterior dorsal striatum (yellow). Middle, a coronal section showing example virus expression. The coronal section is aligned to the Allen Reference Brain (Materials and methods). Right, behavioral performance change relative to the control trials during photostimulation in the sample, delay, or response epoch. Blue, lick left trials; red, lick right trials. Thin lines, individual mice; thick lines, mean. **p<0.01, ***p<0.001, significant performance change compared to the control trials (bootstrap, Materials and methods). (**G**) Same as (**F**) but for photostimulation in the dorsolateral striatum. (**H**) Same as (**F**) but for photostimulation in the posterior dorsal striatum. (**I–L**) Same as (**A–D**) but for photoinhibition in the left superior colliculus. (**M**) Same as (**E**) but for photoinhibition in the left superior colliculus. (**N**) The 3D rendered brain shows the

*Figure 8 continued on next page*

*Figure 8 continued*

striatum and superior colliculus (blue) and the centers of virus expression in individual mice used in this study (dots). See individual mouse data in
*Figure 8—figure supplement 1*.

The online version of this article includes the following figure supplement(s) for figure 8:

**Figure supplement 1.** Virus injection sites in the striatum and superior colliculus.

induced by photoinhibiting the left ALM (*Figure 7G*). This suggests that both ALM and SC are involved in the tactile decision task during the delay epoch.

These experiments show that the automated workflow could be used to rapidly survey distributed brain networks involved in behavior, including deep brain regions (*Figure 8N*).

## Discussion

### Fully autonomous home-cage mouse behavioral and optogenetic experiments

We present a fully autonomous workflow for high-throughput mouse behavioral and optogenetic experiments (*Figure 1*). Mice engaged in voluntary head-fixation in an autonomous home-cage system that was amenable to operant conditioning (*Figure 2*). We developed algorithms that trained completely naive mice to perform tactile decision-making without human supervision (*Figure 3*). We integrated a fiber-free optogenetic method to manipulate activity in specific brain regions during home-cage behavior. We characterized the optogenetic approach using electrophysiology and loss-of-function experiments (*Figure 7*). Finally, we show that the workflow can be used to rapidly survey subregions of the striatum and downstream superior colliculus involved in decision-making (*Figure 8*).

Our approach presents three key advances. First, we introduce a low-cost, open source, and robust home-cage system that allows continuous task training (>1 hr per day) for 2 months without human supervision. Our system significantly boosts the yield and duration of home-cage training to rival and slightly surpass that of manual training (*Supplementary file 1*). This lowers the barrier for training mice in difficult operant conditioning tasks. We show that mice in home-cage training robustly learned a tactile decision task with short-term memory, and they robustly learned contingency reversals in which they flexibly reported decisions using directional licking (*Figure 5*). These tasks are previously difficult to train and require human expertise. Manual behavioral training is often not well documented. The automation and standardization afforded by the home-cage system increase the ease of transferring behavioral paradigms across labs.

Second, we provide the first benchmark dataset that shows fully automated experiments could supersede manual experiments. We show that automated training has similar success rate and speed as manual training (*Figure 3*). A logistic regression model of the choice behavior shows that mice in home-cage training learned the task using similar strategies as in human-supervised training (*Figure 4*). The behaviors resulting from home-cage training engaged the same cortical regions as manual training (*Figure 7*; *O'Connor et al., 2013*; *Sachidhanandam et al., 2013*; *Guo et al., 2014a*; *Li et al., 2015*). In addition, we directly demonstrate the capacity for high-throughput experiments by testing dozens of mice at a time in parallel.

Finally, our workflow is the first to combine home-cage training and unsupervised optogenetic testing. We provide a fiber-free method to manipulate deep brain regions and provide characterization data to show that the method can potently modulate neural activity and bias behavior (*Figure 8*). Voluntary head-fixation and photostimulation through an intact skull bypasses the need to manually tether the mice to a light source and it facilitates continuous optogenetic testing across days without human interventions. In the fully automated workflow, only one injection and headbar implant surgery is needed to prepare a mouse and little supervision is needed thereafter. Parallel testing allows a large number of mice and brain regions to be tested in a single behavior. The approach will enable rapid surveys of distributed brain networks underlying operant behaviors in mice.

Our workflow is particularly suitable for mapping cortico-basal-ganglia loops involved in operant behaviors that require extended training. The striatum is topographically organized (*Hintiryan et al.,*

*2016*; *Hunnicutt et al., 2016*; *Hooks et al., 2018*; *Peters et al., 2019*; *Lee et al., 2020*). The striatum in the mouse brain is ~21.5 mm$^3$ in size (Allen reference brain, *Wang et al., 2020*). Optogenetic experiments using optical fibers manipulate activity near the fiber tip (approximately 1 mm$^3$). Previous studies examined different subregions of the striatum in different perceptual decision behaviors, making comparisons across studies difficult. A systematic survey of different striatal domains' involvement in specific behaviors is currently difficult. In our workflow, individual striatal subregions (~1 mm$^3$, *Figure 8*) could be rapidly screened through parallel testing. At moderate throughput (15 mice/2 months), a screen that tiles the entire striatum could be completed in under 12 months with little human effort. To illustrate its feasibility, we tested three subregions in the striatum previously implicated in different types of perceptual decision behaviors (*Znamenskiy and Zador, 2013*; *Sippy et al., 2015*; *Yartsev et al., 2018*). The results show that our approach could reliably differentiate striatal regions that biased tactile decision-making from those that did not (*Figure 8F–H*).

By eliminating human intervention, automated training also allows quantitative assaying of task learning (*Figure 4*). Home-cage testing also exposes behavioral signatures of motivation in self-initiated behavior (*Figure 6*). These observations suggest additional opportunities for inquires of goal-directed behaviors in the context of home-cage testing.

## Relation to previous automated behavioral experiments

Several recent studies have developed automated systems to train rodents in decision-making and motor control tasks (*Erlich et al., 2011*; *Poddar et al., 2013*; *Aoki et al., 2017*; *Silasi et al., 2018*; *Bollu et al., 2019*; *Erskine et al., 2019*; *Qiao et al., 2019*; *Reinert et al., 2019*; *Bernhard et al., 2020*). Automated systems have also been incorporated into large environments to probe social and environmental factors on cognitive behaviors (*Freund et al., 2013*; *Castelhano-Carlos et al., 2014*; *Torquet et al., 2018*). For operant behaviors, automated testing has been combined with imaging (*Scott et al., 2013*; *Murphy et al., 2016*; *Murphy et al., 2020*), lesion (*Kawai et al., 2015*), video-based behavioral analysis (*Qiao et al., 2019*), and optogenetics (*Bollu et al., 2019*). Some of these systems also implement automated head-fixation (*Kampff et al., 2010*; *Scott et al., 2013*; *Murphy et al., 2016*; *Aoki et al., 2017*; *Murphy et al., 2020*). However, most previous systems still require manual interventions to couple the neurophysiology or manipulation apparatus to the animals before each session (*Aoki et al., 2017*; *Bollu et al., 2019*), but see *Scott et al., 2013*; *Murphy et al., 2020*. Moreover, previous home-cage training with head-fixation is limited to relatively simple behavioral tasks and short training durations (*Aoki et al., 2017*; *Murphy et al., 2020*; *Supplementary file 1*). In our workflow, mice can engage in prolonged head-fixation (>1 hour/day for 2 months) that permits extended training (tens of thousands of trials) in difficult behavioral tasks and continuous optogenetic testing in home-cage. Stable head-fixation also makes our workflow compatible with widefield imaging, and potentially two-photon imaging.

Our general approach and workflow are not restrictive to any specific behavioral system. We integrate and validate several design elements from previous studies. For example, our system has a layout design similar to *Murphy et al., 2016*; *Silasi et al., 2018*, where the headport is integrated into the home-cage for easy access. Our head-fixation mechanism is modeled after Scott and Tank (*Scott et al., 2013*). We employ a load cell to measure mice's body weight, based on *Noorshams et al., 2017*. In turn, our automated training protocols (*Figures 2–3*) can be readily used for other behavioral tasks. Importantly, we find that the self-release mechanism is critical for mice to learn voluntary head-fixation (*Figure 2H*). Without it, mice will start to struggle beyond a certain duration, and if failed to get free, mice will stop engaging in head-fixation subsequently. In addition, we find that auto-assistance to the mice is critical for successful task learning (*Figure 3C*). These guidelines will likely generalize to other automated training.

Other design choices not explored here may further improve the efficiency of automated training. Mice in our study are singly housed. Other studies testing group housed mice suggest a potential for higher yield in trial count (*Murphy et al., 2016*; *Reinert et al., 2019*). One factor that may negatively affect yield in group housed mice is social hierarchy. Dominant mouse may occupy the headport most of the time, which could reduce training time for other co-housed mice (*Murphy et al., 2016*). This problem can be alleviated by building behavioral test chambers that are separated from the home-cage (*Castelhano-Carlos et al., 2014*; *Aoki et al., 2017*; *Torquet et al., 2018*; *Qiao et al., 2019*). Access to the test chamber can then be managed using intelligent protocols

based on RFID tags of individual mice (*Lewejohann et al., 2009*; *Bolaños et al., 2017*; *Erskine et al., 2019*).

## Probing brain regions involved in perceptual decision-making

Our optogenetic experiments suggest that a subregion of the dorsolateral striatum and a lateral region of the superior colliculus (SC) are required for tactile-guided licking decisions (*Figure 7N*). These regions overlap with regions in the striatum and SC previously implicated in licking motor control (*Rossi et al., 2016*; *Lee et al., 2020*). In particular, the subregion of the dorsolateral striatum targeted here (*Figure 8—figure supplement 1B*) is slightly dorsal to but has substantial overlap with a ventrolateral region of the striatum that receives strong ALM input (*Hooks et al., 2018*; *Lee et al., 2020*). Stimulation of the ventrolateral striatal region can evoke contralateral licking (*Lee et al., 2020*). It is worth noting that the subregion of the dorsolateral striatum targeted here also receives some ALM input, but it additionally receives input from the barrel cortex (*Sippy et al., 2015*; *Hooks et al., 2018*) and perturbation of this region impairs tactile-guided licking decisions (*Figure 8G*, sample epoch) (*Sippy et al., 2015*). However, our data cannot yet resolve whether the behavioral effects observed here was due to perturbations of part of the ventrolateral striatal region. Perturbations of the anterior and posterior striatum produced small but significant effects (*Figure 8F and G*). The effect was only observed during the delay epoch and only at the highest laser power (*Figure 7—figure supplement 2*). The effect could result from overlaps of the perturbed regions with the dorsolateral striatum. A more systematic mapping around these striatal regions is needed to determine whether a discrete subregion of the striatum contributes to licking decisions. Our high-throughput workflow is ideally suited for such survey studies.

One limitation of the current workflow is the interpretation of deficit effect size induced by photostimulation. In previous studies, we have shown that photoinhibition of ALM results in chance level performance (*Li et al., 2016*; *Gao et al., 2018*). In this study, mice performance was above chance during photoinhibition of ALM (*Figure 7—figure supplement 1A*). This difference in effect size likely resulted from incomplete silencing of ALM. The photostimulus intensity used here was less than those used in previous studies (*Figure 7—figure supplement 1*). In addition, a single virus injection was not sufficient to cover the entire ALM (1 mm$^2$ in diameter) (*Chen et al., 2017*). Thus a partial behavioral effect could be due to incomplete silencing of a brain region, or partial involvement of the brain region in the task.

Given this limitation, manipulations alone cannot yet elucidate the function of a brain region in behavior. The workflow presented here can be used as a discovery platform to quickly identify regions of interest for more detailed neurophysiology analysis. Our proof-of-concept experiments show that our automated workflow can be a useful tool to facilitate discovery of distributed multiregional networks driving complex behaviors, and it paves the way for more targeted neurophysiology analysis.

# Materials and methods

**Key resources table**

| Reagent type (species) or resource | Designation | Source or reference | Identifiers | Additional information |
|---|---|---|---|---|
| Strain, strain background (Mouse) | Gad2-IRES-Cre | The Jackson Laboratory | JAX: 010802 (RRID:IMSR_JAX:014548) | Cre targeted at the *Gad2* locus |
| Strain, strain background (Mouse) | PV-IRES-Cre | The Jackson Laboratory | JAX: 008069 (RRID:IMSR_JAX:008069) | Cre targeted at the *Pvalb* locus |
| Strain, strain background (Mouse) | VGAT-ChR2-EYFP | The Jackson Laboratory | JAX: 014548 (RRID:IMSR_JAX:014548) | ChR2 targeted at the *Slc32a1* locus |
| Strain, strain background (Mouse) | Ai32(RC-hR2(H134R)/EYFP) | The Jackson Laboratory | JAX: 012569 (RRID:IMSR_JAX:012569) | ChR2 targeted at the *Gt(ROSA)26Sor* locus |
| Recombinant DNA reagent | AAV9-hSyn-FLEX-Chrimson R-tdTomato | UNC Viral Core | N/A | |

*Continued on next page*

*Continued*

| Reagent type (species) or resource | Designation | Source or reference | Identifiers | Additional information |
|---|---|---|---|---|
| Recombinant DNA reagent | AAV8-Ef1a-DIO-ChRmine-mScarlet-WPRE | Stanford Viral Core | GVVC-AAV-188 | |
| Recombinant DNA reagent | AAV-pCAG-FLEX-EGFP-WPRE | Addgene | 51502 (RRID:Addgene_51502) | |
| Software, algorithm | MATLAB | Mathworks | https://www.mathworks.com | |
| Other | Design files, software, and documentations for the automated home-cage system. | This paper - Github repository | https://github.com/ NuoLiLabBCM/Autocage | The Github repository contains the hardware design files and software for the construction of automated home-cage system, along with documentations and protocols for automated head-fixation training and task training. |

## Hardware and code availability

All hardware design files and software for the construction of automated home-cage system are made available, along with documentations and protocols for automated head-fixation training and task training (https://github.com/NuoLiLabBCM/Autocage). Behavioral data and code used are also available in the Github repository.

## Mice

This study is based on data from 140 mice (both males and females, 2–6 months old). 50 GAD2-IRES-Cre (Cre targeted at the *Gad2* locus, Jackson Laboratory, JAX 010802) and PV-IRES-Cre (Cre targeted at the *Pvalb* locus, JAX 008069) mice were used for automated home-cage training and optogenetic experiments targeting the barrel cortex, ALM, striatum and superior colliculus. eight mice were used for contingency reversal learning experiment. Another three GAD2-IRES-Cre mice were used for control optogenetic experiments in which GFP viruses were injected into ALM. An additional four mice were trained in home-cage without self-release mechanism. Another five GAD2-IRES-Cre mice were used for electrophysiology to characterize the effects of optogenetic manipulation in the barrel cortex (two mice), striatum (two mice), and superior colliculus (one mouse). Seventy mice, including GAD2-IRES-Cre, VGAT-ChR2-eYFP (ChR2 targeted at the *Slc32a1* locus, JAX 014548), Ai32 (ChR2 targeted at the *Gt(ROSA)26Sor* locus, Rosa26-LSL-ChR2-eYFP, JAX 012569), and wild-type mice, were used for supervised manual training.

All procedures were performed in accordance with protocols approved by the Institutional Animal Care and Use Committees at Baylor College of Medicine (protocol AN7012). Mice were housed in a 12:12 reverse light:dark cycle. On days not tested, mice received 0.5–1 ml of water. In home-cage experiments, mice were singly housed in the automated home-cage 24/7 and received unrestricted access to the lickport and water rewards by engaging in the task. Body weights were monitored daily. Mice experiencing loss of body weight or failed to engage in voluntary head-fixation for prolonged time period were removed from the study (2/61 mice). In manual training, mice were tested in daily sessions during the dark phase. Experimental sessions lasted 1–2 hr, during which mice received all their water (0.3 to 2 ml). On days not tested, mice received 0.5–1 mL of water. In all cases, if mice did not maintain a stable body weight, they received supplementary water (*Guo et al., 2014b*).

## Headbar implant surgery and virus injection

All surgical procedures were carried out aseptically under 1–2% isoflurane anesthesia. Buprenorphine Sustained Release (1 mg/kg) and Meloxicam Sustained Release (4 mg/kg) were used for pre- and post-operative analgesia. After the surgery, mice were allowed to recover for at least 3 days with free access to water before water restriction.

Mice were prepared with a clear-skull implant and a custom headpost (*Figure 7A*; *Guo et al., 2014a*). In brief, the scalp and periosteum over the dorsal surface of the skull were removed. A layer of cyanoacrylate adhesive (Krazy Glue, Elmer's Products) was directly applied to the intact skull. A custom-made headpost (*Figure 2A*) was placed on the skull (approximately over the cerebellum)

and cemented in place with clear dental acrylic (1223CLR, Lang Dental). A thin layer of clear dental acrylic was applied over the cyanoacrylate adhesive covering the entire exposed skull.

In GAD2-IRES-Cre and PV-IRES-Cre mice prepared for optogenetic experiments, a small craniotomy (~0.5 mm diameter) was made to inject viruses through a pulled glass pipette (20–30 μm tip diameter). For photoinhibition of cortical regions, we injected 200 nL of AAV9-hSyn-FLEX-ChrimsonR-tdTomato (UNC viral core, titer 5.7 × $10^{12}$ vg/ml) (*Klapoetke et al., 2014*) in PV-IRES-Cre or GAD2-IRES-Cre mice (*Figure 7*). The injection coordinates were as follows: the left or right ALM, anterior 2.5 mm from bregma, lateral 1.5 mm, depth 0.4 and 0.8 mm; the left somatosensory cortex, posterior 1.5 mm from bregma, lateral 3.5 mm, depth 0.4 and 0.8 mm. For injections in the somatosensory cortex, we injected viruses in three adjacent sites (~0.5 mm apart) to cover the whole barrel cortex. For photoinhibition of the lateral superior colliculus, we injected 200 nL of AAV8-Ef1a-DIO-ChRmine-mScarlet-WPRE (Stanford viral core [*Marshel et al., 2019*], titer 3.3 × $10^{13}$ vg/ml) or AAV9-hSyn-FLEX-ChrimsonR-tdTomato in GAD2-IRES-Cre mice at posterior 3.5 mm from bregma, lateral 1.5 mm, depth 2.4 and 2.6 mm. For photostimulation of the striatal regions, we injected 100 nL of AAV8-Ef1a-DIO-ChRmine-mScarlet-WPRE in the left anterior dorsal striatum (anterior 1.3 mm from bregma, lateral 1.8 mm, depth 2.2 mm), or the left dorsolateral striatum (anterior 0.6 mm from bregma, lateral 2.4 mm, depth 2.2 mm), or the left posterior dorsal striatum (posterior 0.4 mm from bregma, lateral 2.0 mm, depth 2.1 mm) of GAD2-IRES-Cre mice. For control experiments, we injected 100 nL AAV-pCAG-FLEX-EGFP-WPRE (AddGene, 51502) into the left ALM of 2 GAD2-Cre mice.

## Autonomous home-cage system hardware
### Behavioral test chamber
The core of the autonomous home-cage system was a behavioral test chamber attached to the mouse home-cage. An 'L'-shape board (180 × 100 × 72 mm) housed all components of the behavioral test chamber as shown in *Figure 1C*. The board was designed using CAD software (Fusion 360, Autodesk) and 3D-printed with Nylon material. The board has a headport (~20 × 20 mm) in the center that accessed the home-cage. The board was attached to a standard mouse cage (290 × 180 × 120 mm) using two screws. A 25 × 25 mm pass-through was made on wall of the mouse cage to connect with the headport.

The two sides of the headport were fitted with widened tracks that guided a custom headbar (26.5 mm long, 3.2 mm wide) into a narrow spacing where the headbar could be clamped (*Figure 2B*). Two snap action switches (D429-R1ML-G2, Mouser) were mounted on both sides of the headport. The first 3 mm of the switch tips were bent 90 degrees to fit into the slots of the headport to detect headbar entries (*Figure 2B*). Two air pistons (6604K11, McMaster), with tips processed into a cone shape, were fixed into two holes above the headport. The pistons were pneumatically driven and controlled by an analog pressure regulator (557773, Festo).

A lickport with two lickspouts (5 mm apart) was placed in front of the headport. The lickport was actuated by two orthogonally fixed miniature linear motors (L12-50-100-12-I and L12-30-50-12-I, Actuonix), one moving the lickport forward and backward (i.e. toward or away from the headport) and the other in the left and right directions. Each of the lickspout was electrically coupled to a custom circuit board that detected licks via completion of an electrical circuit upon licking contacts (*Slotnick, 2009*; *Guo et al., 2014a*). Water rewards were dispensed by two solenoid valves (LHDA1233215H, Lee Co.).

The sensory stimulus for the tactile decision task was a mechanical pole (1.5 mm diameter) on the right side of the headport. The pole was motorized by a linear motor (L12-30-50-12-I, Actuonix) and presented at different location to stimulate the whiskers. The motorized pole was attached to an air piston (6498K999, McMaster), driven by a 3/2-way solenoid valve (196847, Festo), which moved the pole vertically into the reach of the whiskers. The entire pole mechanism was mounted on a holder on the behavioral test chamber board. The auditory 'go' cue in the tactile decision task was provided by a piezo buzzer (3.5 kHz, cpe163, Mouser) placed in front of the headport.

A custom 3D-printed platform was placed inside the home-cage in front of the headport (*Figure 1C*). The stage was embedded with a load cell (*Figure 2C*, CZL639HD, Phidgets) to record mouse body weight. The load-sensing stage was also used to detect struggles during head-fixations and trigger self-release (*Figure 2D*). The surface of the stage was coated with aluminum foil to

produce electrical contact between the mouse and the electric lickspouts upon licking. The aluminum foil is connected to the lick detection circuit board.

An optical fiber (M79L005, Thorlabs), coupled to a red laser (633 nm, MRL-III-633–50, Ultralaser) or a LED (625 nm, M625F2, Thorlabs), was mounted above the headport. The fiber was approximately 12 mm above the clear skull implant during head-fixations. In cases where higher light power was needed, the optical fiber was placed closer to the mouse and aimed at the photostimulation site. To prevent mice from distinguishing photostimulation trials from control trials using visual cues, a masking flash (10 Hz) was delivered using a 627 nm LED (SP-01-R5, LexonStar) mounted in front of the headport.

## Controllers

Three Arduino microcontrollers (A000062, Mouser) operated the home-cage system (*Figure 1—figure supplement 1A*). A 'master' controller stored the protocols for head-fixation training (*Figure 2E*), task training (*Figure 3C*), and optogenetic experiments (see *Autonomous training protocols*). The master controller autonomously advanced these protocols based on mouse behavioral performance. In addition, the master controller controlled the head-fixation logics, the lickport motors, and the motor that positioned the pole (i.e. tactile stimulus). For head-fixations, the controller read switch triggers through a digital input/output (DIO) port and controlled the pressure regulator that actuated the pneumatic pistons via a digital-to-analog converter (DAC) port. For the self-release mechanism, the output from the load cell was amplified (HX711, SparkFun) and read by the master controller through a DIO port (sampled at 20 Hz). To position the lickport and the pole, the master controller interfaced with the motors through DIO ports.

A second 'task' controller controlled individual behavioral trials using finite state machines with high temporal resolution (0.1 ms, adapted from the open-source Bpod project https://github.com/sanworks/Bpod_StateMachine_Firmware; *Sanders, 2018*). The task controller was triggered by the master controller. Before each trial, the master controller generated a finite state machine and sent the state matrix to the task controller to execute. The task controller actuated the air piston that presented the pole and the solenoid valves that delivered reward through solid state relays (DMO063, Mouser). The task controller also actuated the piezo buzzer for the 'go' cue via a pulse width modulation (PWM) port. The task controller read licking events (TTL high logic) from the lick detection circuit board.

Finally, a third 'wave' controller was used to generate custom waveforms to drive the optogenetic components, including the masking flash LED (through an LED driver, SS25S075, SparkFun) and a red laser (through a high-precision DAC, MCP4725, Adafruit). The 'wave' controller received meta information from the master controller that specified the output waveform (e.g. amplitude) before each trial, but the output was triggered by the task controller during specific task epochs.

The home-cage system operated standalone. The master controller was equipped with a data logging shield (ID1141, Adafruit) and a real time clock (RTC) module to timestamp and store all behavioral data and training parameters on a SD card. Behavioral data included detailed information for each trial (e.g. trial number, trial type, trial outcome, licking events, etc.) and head-fixation events (e.g. switch trigger, head-fixation, release, etc.). Training parameters specified the current protocol and training progression. Behavioral data and training parameters were updated after each behavioral trial. Each mouse was associated with its own SD card that contained its behavioral data and training parameters. If interrupted, the home-cage system could resume training based on the training parameters stored on the SD card. In this manner, each mouse could switch between any home-cage systems using its unique SD card.

## Enclosure

The entire behavioral system was fit into an enclosure (560 × 250 × 230 mm) constructed with rails (XE25L, Thorlabs) and detachable acrylic boards (8505K745, McMaster). The top board of the enclosure was cut with an opening (170 × 170 mm) to provide light access during light cycles. The opening was protected with a mesh screen. The enclosed system was standalone and powered by a 12 V power supply (for Arduino microcontrollers), a 24 V power supply (for solenoid valve and analog pressure regulator), a 4 bar air supply (for the pneumatics and analog pressure regulator). In addition, each system has an optional USB cable (from the master controller) to stream real-time data for

display on a PC (*Figure 1—figure supplement 1*). However, the system could operate without the PC display. Multiple systems could be connected to a single PC through a USB hub (*Figure 1—figure supplement 1*).

## Autonomous training and optogenetic testing protocols

Three separate protocols ('head-fixation training', 'task training' and 'optogenetics') on the master controller autonomously trained mice in voluntary head-fixation and the tactile decision-making task, as well as carrying out optogenetic testing.

### Head-fixation training protocol

Head-fixation training had two subprotocols. A 'headport entry' subprotocol lured mice into the headport and acclimated them to headport entry. A second 'head-fixation' subprotocol acclimated mice to head immobilization (*Figure 2E*).

The headport entry subprotocol started by placing the lickport close to the headport with the two lickspouts inside the home-cage. Mice could lick both lickspouts freely. However, only licking the rewarded lickspout led to a water reward. Licks on the other lickspout were ignored. The rewarded lickspout alternated between the two lickspouts (three times each) to encourage licking on both lickspouts. This phase of the training acclimated mice to the lickport. After every 20 rewarded licks, the lickport was retracted one step (3 mm) away from the home-cage. The lickport retraction continued until the tip of the lickspouts was approximately 14 mm away from the headport. At this point, mice could only reach the lickspouts by entering the headport with the headbar reaching the end of the guide tracks (*Figure 2B*). This reliably triggered the mechanical switches at the end of the tracks. If mice failed to trigger the switches when the lickport was fully retracted, or if no licks were detected in 12 hr, these scenarios typically indicated that mice were not lured into the headport. In these cases, the headport entry subprotocol would re-extend the lickport toward to the home-cage to lure mice in again (*Figure 2E*). After the mechanical switches were triggered 30 times, the training advanced to the head-fixation subprotocol (*Figure 2E*).

The head-fixation subprotocol started by turning on the pneumatic pistons and the head-fixation control logics. Whenever the switches were triggered, the pneumatic pistons were activated to clamp the headbar. The clamps were released under three scenarios: (1) 'time-up release', when head-fixation lasted for a predefined duration; (2) 'escape', when the switches were no longer triggered by the mice, ('escape' occurred when mice quickly pulled out from the headport before the pistons could clamp the headbar); (3) 'self-release', when the weight readings from the load-sensing platform exceeded a threshold. Head-fixations reduced the weight load on the platform, and overt movements of the mouse typically produced large fluctuations in weight readings (*Figure 2D*). We set two thresholds at $-1$ and 30 g to robustly detect any struggles, that is whenever the weight readings fell below $-1$ g or exceeded 30 g, the clamps were released. These thresholds were dynamically adjusted during the training process: if there were too many self-releases (>90% of the head-fixations), the upper threshold would increase by 2 g and the lower threshold would decrease by 2 g (increasing the range); conversely, if there were too few self-releases (<5%), the upper threshold would decrease by 2 g and the lower threshold would increase by 2 g (decreasing the range). The thresholds were adjusted based on the last 20 head-fixations.

Initially, head-fixation started with a 'soft clamp' mode (pistons pressure 1.78 bar) and each head-fixation lasted for a short duration (time-up release, 3 s). The duration was increased by 2 s after every 20 successfully head-fixations (i.e., time-up release). After the duration reached 10 s, head-fixation switched to a 'hard clamp' mode. In the hard clamp mode, each head-fixation started with low pressure (1.78 bar), but the pressure increased to 2.78 bar after the first 2 s of fixation. At the end of each head-fixation, if the mouse did not pull out from the headport after the pistons were release (i. e., the switches remained triggered), the next head-fixation would initiate. The head-fixation training subprotocol completed when the fixation duration reached 30 s (*Figure 2E*), and the master controller automatically advanced to the task training protocols. From this point onward, the head-fixation control logics ran continuously.

Importantly, we found it necessary to acclimate mice with task stimuli during head-fixation training, well before task training started (see below). Introducing novel stimuli after head-fixation training often caused an increase in self-release rate and reduced the number of head-fixations, which

hindered task training. To circumvent this issue, we introduced the tactile stimulus and the auditory 'go' cue at the beginning of the headport entry subprotocol. Specifically, the pole was presented to touch the whiskers at the locations corresponding to the rewarded lickspout. Water reward was only given for the first lick on the rewarded lickspout after the 'go' cue sound.

## Task training protocol

We trained mice in a tactile decision task with a short-term memory component. Mice used their whiskers to discriminate the location of a pole and reported choice using directional licking for a water reward (*Figure 3A*, anterior pole position→lick left, posterior pole position→lick right) (*Guo et al., 2014b*; *Guo et al., 2014a*). The pole was presented at one of two positions that were 6 mm apart along the anterior-posterior axis. The posterior pole position was approximately 5 mm from the whisker pad. The pole was always presented to the right whiskers. In each head-fixation, mice were tested in a succession of trials. At the beginning of each trial, the pole moved into reach of the whiskers (0.2 s travel time), where it remained for 1 s, after which it was retracted (retraction time 0.2 s). The sample epoch was defined as the time between the pole movement onset to 0.1 s after the pole retraction onset (sample epoch, 1.3 s). A delay epoch followed, during which the mice must keep the information in short-term memory (delay epoch, 1.3 s). An auditory 'go' cue (pure tone, 3.5 kHz, 0.1 s duration) signaled the beginning of response epoch and mice reported choice by licking one of the two lickspouts. Each trial was followed by an inter-trial-interval (2.5 s), after which the next trial began, until the head-fixation is released (*Video 1*).

Task training had three subprotocols that shaped mice behavior in stages (*Figure 3C*). First, a 'directional licking' subprotocol trained mice to lick both lickspouts and switch between the two. Then, a 'discrimination' subprotocol taught mice to report pole position with directional licking. Finally, a 'delay' subprotocol taught mice to withhold licking during the delay epoch and initiate licking upon the 'go' cue.

The directional licking subprotocol started immediately after mice completed the head-fixation training. Lick left or lick right trials were presented consecutively. Mice had to obtain three trials correct before the program switched to the other trial type. At this stage of the training, the pole was presented for 1.3 s and a short delay epoch (200 ms) was included but not enforced. Mice were free to lick at any time during the trial, but only the first lick after the 'go' cue were registered as choice. Licking the correct lickspout after the 'go' cue led to water reward (2–3 µL). Licking the incorrect lickspout triggered a timeout (2 s). Trials in which mice did not lick within a 1.5 s window after the 'go' cue were counted as ignores.

The discrimination subprotocol started once mice reached 70% correct in the directional licking subprotocol (assessed over 30 trials). All aspects of the task remained the same, but the lick left and lick right trials were presented in random order. Several auto-assist programs tracked mice's performance and occasionally adjusted the probability of each trial type (see *Auto-assist programs*). In addition, licking the incorrect lickspout triggered a longer timeout (4 s).

The delay subprotocol started once mice reached 75% correct in the discrimination subprotocol. At this stage, licking before the 'go' cue triggered a brief pause (0.1 s). After the pause, the program resumed the trial from the beginning of the epoch in which the early lick occurred (sample or delay). This constituted an additional timeout. However, the mouse could still complete the trial to obtain a reward if it licked the correct lickspout after the 'go' cue. Initially, the delay epoch was brief (0.3 s). The duration of the delay epoch increased by 0.2 s every time mice reached 70% correct performance. The delay subprotocol ended when the delay epoch duration reached 1.3 s.

At the end of the delay subprotocol, the head-fixation duration was further increased from 30 to 60 s. The duration was increased by 2 s after every 20 successfully head-fixations. This was done to obtain more behavioral trials in each head-fixation.

## Auto-assist programs

During task training, mice often developed idiosyncratic biases by licking one lickspout more frequently or continuously licking one lickspout without switching to the other. We implemented four different 'auto-assist' programs to counter these behavioral patterns (*Figure 3C*). These auto-assist programs evaluated mice performance continuously and assisted mice whenever behavioral biases were detected.

First, if a bias developed (i.e. performance difference between the two trial types exceeded 30% in the last 50 trials or exceeded 80% in the last 20 trials), the left/right motor moved the lickport such that the non-preferred lickspout was closer to the mouse. Second, if a mouse made five consecutive errors in one trial type, a free water reward was delivered to the rewarded lickspout in the next trial. This motivated the mouse to lick the rewarded lickspout. Third, the program calculated behavioral performance in the last 30 trials and the trial type with worse performance was presented more frequently. Finally, if a mouse made three consecutive errors in one trial type, the program would keep presenting the same trial type until the mouse got two trials correct.

## Contingency reversal learning

Mice were trained in the tactile decision task without a delay epoch. The training protocol is the same as described above, with the exception that the delay subprotocol was not included. Mice always learned the standard contingency first where posterior pole position corresponded to lick right and anterior pole position corresponded to lick left. Once performance was >80% correct for 100 trials, the correspondence between pole locations and lick directions were reversed. Mice did not receive any cues about the reversal other than reward feedbacks: correct responses led to rewards and incorrect responses led to timeouts. The new contingency remained in place until mice reached 80% correct, upon which the continency was reversed again.

## Optogenetics protocol

The optogenetics protocol was manually initiated by experimenters based on inspections of behavioral performance (*Figure 1—figure supplement 1B*). In the optogenetics protocol, photostimulation was given in a random subset of trials (10%) during the sample, delay, or response epoch. Photostimulation power was randomly selected (see *Optogenetic experiments*). In addition, a masking flash (10 Hz) was given on all trials. Masking flash began at the start of the sample epoch and continued through the end of the response epoch in which photostimulation could occur.

## Measuring head-fixation stability

To measure the repositioning of the headbar across multiple head-fixations, we used a CMOS camera (CM3-U3-13Y3M, FLIR) to measure the displacements of the clear skull implant. To measure the displacements in the rostral-caudal and medial-lateral directions, the camera was placed over the headport (pixel resolution, 52.6 μm/pixel). To measure the displacement in the dorsal-ventral direction, we glued a small plastic marker on the clear skull implant and placed the camera in front of the headport (pixel resolution, 78.1 μm/pixel) to measure the vertical displacement of the marker. Across 16 different head-fixations, 10 images were acquired at random time points during each head-fixation.

To determine the displacement, we selected small regions of interest (ROIs, 30 × 30 pixels) on the clear skull implant or the marker (*Figure 2—figure supplement 1*). For each ROI, we computed 2D cross-correlations for every possible image pairs across the 16 different head-fixations (*Figure 2—figure supplement 1A*). Cross-correlation (*xcorr2* function, MATLAB) always produced zero-pixel shift, indicating that any displacement was at a subpixel scale. We thus employed a subpixel image registration algorithm (*Guizar-Sicairos et al., 2008*), which measured 2D rigid translation for a small fraction of pixels to calculate the displacements. Displacements in the rostral-caudal/medial-lateral directions and the dorsal-ventral direction were quantified separately since it involved different camera configurations. Displacements were calculated for multiple ROIs (*Figure 2—figure supplement 1A–F*). *Figure 2I* shows the displacements pooled across all ROIs (mean, 8.8, 6.4, and 12.1 μm in rostral-caudal, medial-lateral, and dorsal-ventral direction). The mean and SD were calculated on the absolute values of the displacement.

To verify that the subpixel algorithm captured the displacements accurately, we also selected a ROI on the wall of the headport. The ROI on the wall of the headport showed little displacement (*Figure 2—figure supplement 1G*; mean, 2.07 and 1.10 μm in rostral-caudal/medial-lateral directions). We also selected a ROI containing the mouse's whiskers. The ROI on whiskers showed large displacement (*Figure 2—figure supplement 1H*; mean, 266 and 351 μm in rostral-caudal/medial-lateral directions).

## Manual behavioral training

The procedures for manual behavioral training have been described previously (*Guo et al., 2014b*). Mice were manually acclimated to head-fixation and tested in daily sessions that lasted 1–2 hr. The training started by rewarding mice for simply licking the lickspouts. The auditory 'go' cue was played immediately before water delivery.

After mice learned to lick for water (~60 rewards), the reward scheme was changed to teach mice to sample both lickspouts (similar to the directional licking training protocol in home-cage training). Only one lickspout held a water reward and the rewarded lickspout alternated after three rewards. Occasionally, manual water delivery was necessary to prompt mice to lick from the other lickport. In addition, the rewarded lickspout was moved closer to the mouse in each trial to encourage licking. Gradually, the movement of the lickport was reduced and the lickport eventually remained in a fixed center position. During this phase of the training, the vertical pole was also presented at the position corresponding to the rewarded lickport (anterior pole position→lick left, posterior pole position→lick right). Presentation of the pole allowed mice to gradually associate the pole position with the rewarded lickspout.

Once mice reliably switched licking between lickspouts, object location discrimination task started (equivalent to the discrimination training protocol in home-cage training). In this stage of the training, the two trial types were presented in random order and the task did not include a delay epoch. Mice were free to lick at any time during the trial, but only the first lick after the 'go' cue were registered as choice. Correct choice led to a water reward. Incorrect choice led to a time out.

After mice reached performance criterion (>70% correct), the delay epoch was introduced. Licking before the 'go' cue triggered an alarm sound from a siren buzzer and a brief timeout. The delay epoch was initially short (0.3 s) and gradually increased to 1.3 s.

## Optogenetic experiments

For mice tested in unsupervised optogenetic experiments (see *Optogenetics protocol*), light from a 633 nm laser (MRL-III-633–50, Ultralaser) or 625 nm LED (M625F2, Thorlabs) was delivered via an optical fiber (M79L005, Thorlabs) above the headport. The photostimulus was a 40 Hz sinusoid lasting for 1.3 s, including a 100 ms linear ramp during photostimulus offset to reduce rebound neuronal activity (*Guo et al., 2014a*; *Li et al., 2019*). Photostimulation started at the beginning of the sample, delay, or response epoch. We used average power of 5–50 mW at the fiber tips, which corresponded to light intensity of 0.3–3.5 mW/mm$^2$ given the size of the light beam on the surface of the skull.

For mice tested in photoinhibition of vS1 and ALM, we subsequently tested them in manual optogenetic experiments. In daily sessions, mice were manually head-fixed and tested by an experimenter on an electrophysiology setup. The light source and light delivery were the same as the home-cage optogenetic experiments. The size of the light beam on the brain surface was also matched. To prevent the mice from distinguishing photostimulation trials from control trials using visual cues, a masking flash was delivered using 627 nm LEDs (Luxeon Star) near the eyes of the mice. The masking flash began at the start of the sample epoch and continued through the end of the response epoch in which photostimulation could occur.

## Behavioral data analysis

The duration of each head-fixation was calculated as the interval between the onset of the piston clamps and the following release (*Figure 2*). The inter-fixation-interval was calculated as the interval between a release and the next head-fixation (*Figures 2L* and *6*). Performance was computed as the percentage of correct choices. Mice that never exceeded 70% correct after 35–40 days of training were deemed as unsuccessful in task training. To quantify the effect of photostimulation, we also separately computed performance for lick left and lick right trials. Chance performance was 50%.

Behavioral effects of photostimulation for each mouse were quantified by comparing its performance under photostimulation with control trials. The within-mouse performance changes were then averaged across mice. Significance of the average performance change in each photostimulation condition was determined using a nested bootstrap to account for variability across mice, sessions, and trials. We tested against the null hypothesis that the average performance change caused by photostimulation was due to normal behavioral variability. In each round of bootstrap, we generated

a resampled dataset by first resampling (with replacement) the mice included in the analysis. We then resampled (with replacement) the sessions of each mouse. For each session, we then resampled (with replacement) the trials in the session. We computed the performance change on the resampled dataset. Repeating this procedure 10,000 times produced a distribution of performance changes that reflected the behavioral variability. The p value of the observed performance change was computed as fraction of times the bootstrap produced an inconsistent performance change (e.g. if a performance decrease was observed during photostimulation, the p value is the fraction of times a performance increase was observed during bootstrap, one-tailed test). In this bootstrap analysis, each day (dark +light cycle) in the home-cage optogenetic experiments was treated as a 'session'.

## Logistic regression model of behavioral data

We used a logistic regression model to predict mice's choice. The probability of choice in the current trial, *P(left)*, was a logistic function of the weighted sum of several behavioral and task variables. The variables are the tactile stimulus in the current trial ($S_0$, 1 for lick left trial, -1 for lick right trial), the tactile stimuli in the last five trials ($S_1$ to $S_5$), choice in the last five trials ($A_1$ to $A_5$, 1 for licking left, -1 for licking right), reward outcomes in the last five trials ($R_1$ to $R_5$, 1 for rewarded, -1 for unrewarded), the average stimuli in the last 20 trials ($S_{avg} = \sum_{i=1}^{20} S_i/20$), a win-stay-lose-switch strategy (*WSLS* = $A_1 \times R_1$), and a constant bias term $\beta_0$. The model was defined by the following equations:

$$P(left) = \frac{1}{1 + e^{-(z)}}$$

$$z = \beta_{s_0} S_0 + \sum_{i=1}^{5} \beta_{s_i} S_i + \sum_{i=1}^{5} \beta_{A_i} A_i + \sum_{i=1}^{5} \beta_{R_i} R_i + \beta_{s_{avg.}} S_{avg.} + \beta_{WSLS} WSLS + \beta_0$$

where the β's were the weights for the regressors. *P(left)*>0.5 predicted licking left and *P(left)*<0.5 predicted licking right.

For each mouse, we built separate logistic regression models at different stages of learning. The behavioral trials were concatenated in time, that is, across multiple head-fixations for home-cage training and across multiple sessions for manual training. We used a sliding window of 500 trials (in 100-trial steps). In each window, the model was fit to the behavioral choice data using a gradient descent algorithm to maximize the likelihood estimation cost function (*glmfit*, MATLAB). The model performance was evaluated using 5-fold cross validation. In each round of cross validation, 60 consecutive trials and their history data (the 20 trials before) were selected as the test set. A total of 400 trials were used as the training set while excluding the 20 trials before and 20 trials after the block of trials used as the test dataset. This ensured absolute independence of the training and test data since the logistic regression model used trial history data as its input. This cross-validation procedure was repeated nine times and the prediction performance was averaged. To quantify the model prediction performance, we computed the fraction of trials in which the model correctly predicted choice in the test dataset.

During home-cage training, mice first underwent the 'directional licking' subprotocol that presented the same trial type in blocks to teach the mice to lick both lickspouts (see *Task training protocol*). This phase of the training could introduce dependencies of choice on the choice history. Thus, the logistic regression analysis was performed only on behavioral data after mice advanced to the 'discrimination' phase of the training in which both trial types were presented randomly. For behavioral data from manual training, the logistic regression analysis started at an equivalent time point (see *Manual behavioral training*).

To quantify the contribution of each regressor, we constructed partial models in which specific regressors were removed from the full model. Specifically, we set the weight (β) of the regressor to zero and determined whether the prediction performance of the partial model was significantly worse than the full model (*Engelhard et al., 2019*). To assess statistical significance, we used bootstrap (*Efron and Tibshirani, 1994*) in which the test dataset was resampled with replacement 1000 times and the p value was computed as the fraction of times in which the partial model produced a better performance than the full model (*Figure 4B*). In *Figure 4E*, if a regressor was significant (p<0.05) in five consecutive time windows (1000 trials), the mouse was deemed to rely on this

regressor during task learning. As a control, we also tested the full model fit to a shuffled dataset (*Figure 4C*). To generate the shuffled dataset, we shuffled the choice across trials while maintaining the stimulus and reward history. Model fitting and testing on the shuffled dataset was computed using the same cross validation procedure described above. To examine whether the tactile stimulus ($S_0$) and choice history ($A_1$) were sufficient to account choice prediction of the full model, we tested a reduced model which only included the two regressors and a constant bias term ($\beta_0$) (*Figure 4C*).

## Electrophysiology

To characterize photoinhibition in cortex, we injected 200 nL of AAV9-hSyn-FLEX-ChrimsonR-tdTomato viruses into the left barrel cortex (bregma posterior 1.5 mm, 3.5 mm lateral, 0.4 and 0.8 mm deep) of GAD2-IRES-Cre mice. To characterize the effect of photostimulation in the striatum, we injected 100 nL of AAV8-Ef1a-DIO-ChRmine-mScarlet-WPRE viruses into the left striatum (bregma posterior 0.6 mm, 2.4 mm lateral, 2.2 mm deep) of GAD2-IRES-Cre mice. To characterize photoinhibition in the superior colliculus, we injected 200 nL of AAV8-Ef1a-DIO-ChRmine-mScarlet-WPRE viruses into the left superior colliculus (bregma posterior 3.5 mm, 1.5 mm lateral, 2.2 mm deep) of GAD2-IRES-Cre mice.

Three weeks after the virus injection, we recorded extracellular spikes using 64-channel silicon probes (H2 probes, Cambridge Neurotech) near the injection site. A small craniotomy (diameter,<1 mm) was made one day before the recording session. During the recording session, the silicon probe was acutely inserted into the brain. To minimize brain movement, a drop of silicone gel (3–4680, Dow Corning) was applied over the craniotomy after the electrode was in the tissue. The tissue was allowed to settle for several minutes before the recording started. For recordings in the barrel cortex, the silicon probe was inserted 0.9–1.11 mm below the brain surface. For recordings in the striatum and superior colliculus, multiple recordings were obtained at a range of depths along a penetration (insertion depths, 2.3–3.5 mm). Recording depth was inferred from manipulator depth and verified with histology (*Figure 8A and I*). The voltage signals were amplified and digitized on an Intan RHD2164 64-Channel Amplifier Board (Intan Technology) at 16 bit, recorded on an Intan RHD2000-Series Amplifier Evaluation System (sampling at 20,000 Hz), and stored for offline analysis.

Photostimulation was performed by directing a laser beam over the surface of the brain (*Guo et al., 2014a*). The light source and light delivery were the same as the home-cage optogenetic experiments. Photostimulation was delivered in approximately 7 s intervals. The mice were awake but not engaged in any task. The power was chosen randomly from a predefined set. We used average power of 0.5, 1.2, 3, 5, 10 and 14.5 mW, which corresponded to light intensity of 0.28, 0.68, 1.7, 2.83, 5.66, 8.21 mW/mm$^2$ given the size of the light beam measured at the surface of the skull.

## Electrophysiology data analysis

The extracellular recording traces were band-pass filtered (300–6,000 Hz). Spike events that exceeded four SDs of the background were subjected to manual spike sorting (*Guo et al., 2014a*). 171 single-units were obtained in the barrel cortex. A total of 348 single-units were obtained in the striatum and 170 single-units were obtained in the superior colliculus.

In cortex, fast-spiking (FS) neurons and pyramidal neurons could be putatively distinguished based on spike waveform. Spike widths were computed as the trough-to-peak interval in the mean spike waveform (*Figure 7C*). Units with spike width <0.55 ms were defined as putative FS neurons (14/171) and units with spike widths > 0.55 ms as putative pyramidal neurons (157/171). In the striatum and superior colliculus, cell types were not distinguished based on spike waveform. Instead, we separately analyzed neurons that were significantly excited or inhibited by photostimulation (*Figure 8D and L*).

For each neuron, we computed spike rates during the photostimulus and a baseline period (500 ms time before photostimulus onset). Significant spike rate change was tested using two-tailed t-test (*Figures 7C*, *8C, E and K*). To quantify the effect size of photostimulation, we calculated a 'relative firing rate'. The spike rates during photostimulation were normalized by dividing the baseline spike rate. The relative firing rate reported the spike rate modulation during photostimulation.

## Histology

Mice were deeply anaesthetized with isoflurane and transcranially perfused with PBS followed by 4% paraformaldehyde (PFA). The brains were removed and post-fixed in 4% PFA for 24 hr before transferring to 30% sucrose. 100 µm coronal sections were cut and imaged on a fluorescence macroscope (Olympus MVX10). We aligned each coronal section to the Allen Mouse Common Coordinate Framework (CCF) (*Wang et al., 2020*) using landmark-based image registration. The registration target was the 10 µm per voxel CCF anatomical template. To align a coronal section, we first manually selected the coronal plane in the anatomical template that best corresponded to the section. We then manually placed control points at corresponding local landmarks in each image. Thirty to fifty control points were placed in a single image. Next, the image was warped to the CCF using an affine transformation followed by a non-rigid transformation using b-splines (*Gao et al., 2018*).

## Acknowledgements

We are grateful to Karl Deisseroth, James Marshel, and Stanford Neurosciences Institute Viral Core for sharing the AAV8-Ef1a-DIO-ChRmine-mScarlet-WPRE virus. We thank Ben Scott, Hidehiko Inagaki, Jia Zhu, and Guang Chen for valuable comments on the manuscript, Kaiwen Wu for pilot work on software development, Kunxun Qian for pilot work on logistic regression analysis, Jing Lin for hardware support, Sri Laasya Tipparaju for animal care and histology, and Weiguo Yang for animal weight data. This work was funded by the Robert and Janice McNair Foundation, Whitehall Foundation, Alfred P Sloan Foundation, Searle Scholars Program, Pew Scholars Program, NIH NS112312, NS104781, NS113110, Simons Collaboration on the Global Brain (543005), and McKnight Foundation.

## Additional information

### Funding

| Funder | Grant reference number | Author |
| --- | --- | --- |
| Robert and Janice McNair Foundation | | Nuo Li |
| Whitehall Foundation | | Nuo Li |
| Alfred P. Sloan Foundation | | Nuo Li |
| Searle Scholars Program | | Nuo Li |
| Pew Charitable Trusts | | Nuo Li |
| National Institutes of Health | NS112312 | Nuo Li |
| National Institutes of Health | NS104781 | Nuo Li |
| National Institutes of Health | NS113110 | Nuo Li |
| Simons Foundation | 543005 | Nuo Li |
| McKnight Endowment Fund for Neuroscience | | Nuo Li |

The funders had no role in study design, data collection and interpretation, or the decision to submit the work for publication.

### Author contributions

Yaoyao Hao, Conceptualization, Resources, Data curation, Software, Formal analysis, Validation, Investigation, Visualization, Methodology, Writing - original draft, Writing - review and editing; Alyse Marian Thomas, Investigation, Writing - review and editing, AMT performed early experiments that identified the region of interest in the superior colliculus; Nuo Li, Conceptualization, Formal analysis, Supervision, Funding acquisition, Validation, Investigation, Visualization, Methodology, Writing - original draft, Project administration, Writing - review and editing

## Author ORCIDs
Nuo Li [ID] https://orcid.org/0000-0002-6613-5018

## Ethics
Animal experimentation: All procedures were performed in accordance with protocols approved by the Institutional Animal Care and Use Committees at Baylor College of Medicine (protocol AN7012). All surgical procedures were carried out aseptically under 1-2% isoflurane anesthesia. Buprenorphine Sustained Release (1 mg/kg) and Meloxicam Sustained Release (4mg/kg) were used for pre- and post-operative analgesia.

## Decision letter and Author response
Decision letter https://doi.org/10.7554/eLife.66112.sa1
Author response https://doi.org/10.7554/eLife.66112.sa2

# Additional files

## Supplementary files
• Supplementary file 1. Comparison with previous automated home-cage training systems with voluntary head-fixation.

• Transparent reporting form

## Data availability
All hardware design files and software for the construction of automated home-cage system are made available, along with documentations and protocols for automated head-fixation training and task training. Source data and code are also available at the same Github repository (https://github.com/NuoLiLabBCM/Autocage).

The following dataset was generated:

| Author(s) | Year | Dataset title | Dataset URL | Database and Identifier |
|---|---|---|---|---|
| Li N | 2021 | Autocage version 1.0: Release for Zenodo. | http://www.doi.org/10.5281/zenodo.4716811 | Zenodo, 10.5281/zenodo.4716811 |

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
