## [Decision Letter]

**Acceptance summary:**

This article details a new open source setup and protocol for automated training of mice to a challenging sensory discrimination task. This tool brings a level of automation (no human intervention) never achieved in a context that also allows targeted manipulation brain areas, in a non-invasive manner. Both these aspects and the potential for combined optical imaging will be extremely useful for the neuroscience community.

**Decision letter after peer review:**

Thank you for submitting your article "Fully autonomous mouse behavioral and optogenetic experiments in home-cage" for consideration by *eLife*. Your article has been reviewed by 3 peer reviewers, including Brice Bathellier as the Reviewing Editor and Reviewer #1, and the evaluation has been overseen by Laura Colgin as the Senior Editor. The following individual involved in review of your submission has agreed to reveal their identity: Philippe Faure (Reviewer #3).

Essential Revisions:

While the referees recognize the solidity of this work, a number of revisions are needed to better highlight its novelties with respect to previous publications on behavior during voluntary head-fixation, to reinforce the description of optogenetic silencing results and to help the reader grasp motivation aspects in this self-initiated behavior.

1. Due to the authors' choice of plotting performance in optogenetics, it is not possible to know if the mice reach chance level during silencing. However, appreciating if the chance level is reached during silencing is important to evaluate if the silencing method leads to complete or partial impairments. The authors should thus also provide plots in which performance during optogenetics is express as the distance to chance level (e.g. 60% correct for a balanced binary task is 10% above chance level). Depending on the outcome of this re-plotting, the author should comment, based on the literature, whether incomplete effects are due to incomplete silencing or due to partial involvement of the target region in the task.

2. The comparison to other methods is important. As it stands, these seem marginal. This should be strengthened. It would be also important to highlight what types of questions, qualitatively, can be answered that are not possible (or difficult) otherwise.

3. The task structure needs to be clarified. The sample period (1.3 sec) is followed by a delay epoch (1.3 sec), an auditory go cue and the response epoch. It is indicated that "Mice were free to respond by licking at any time during the trial, but only the first lick after the 'go' cue were registered as choice". The question is what exactly is a trial ? Is it: (i) a sequence sample-delay – response, with only the first lick taken as a response. That is, 1 head fixation = a trial and is associated with a maximum of 1 water reward (2-3 µL). (ii) a succession of sample-delay -1 response sequences during 30 or 60 sec. That is, 1 head fixation is associated with more than one water reward (2-3 µL). I suspect that the first proposal is the correct one, but this should be clearly stated in the manuscript. Along that line, Figure 3B is confusing. Authors should may be indicate the first lick, the one registered as choice in full color and the additional one as transparent?

4. The authors should provide information about the daily water consumption (2-3 µL per lick number of correct trials per days) and whether it is stable during the protocol. Considering that a mouse drinks about 3-5 mL daily if it weighs more than 30 grams, it would correspond to 2000 correct trials more than 16 h of 30 sec head fixed session, if the above interpretation of a trial is correct.

5. In a paper (Torquet et al. 2018) which describes mouse behavior in an automated T-maze task (left or right to access water versus water + sugar), mice showed a decreased return time after choosing the less-rewarded side. This indicated an increased motivation after a failure, but also the fact that some trial can be associated with low motivation for the reward and engagement in the test just for exploration. Here, the authors showed a distribution of inter-fixation interval, with a long tail. Are these inter-fixation intervals correlated with the success or failure in the last trial which could indicate different motivation depending on the intervals?

6. In the contingency reversals task, do the animals adapted their inter-fixation intervals just after the reversal?

7. The position of the pole with respect to the head (stimulation of the left or right whiskers) is not clearly indicated in the method, information seems to be shown only in Figure 3A. This should be indicated clearly. Along the same lines, on line 498, replace "Photoinhibition of S1" by "Photoinhibition of left S1".

8. The statistics used in the optogenetic experiment is based on bootstrap and unilateral testing (one tail test). It is indicated that "In each round of bootstrap, we replaced the original behavioral dataset with a re-sampled dataset in which we re-sampled with replacement from: (1) mice, (2) sessions performed by each mouse, (3) trials within each session. We then computed the performance change on the re-sampled dataset. Repeating this procedure 10,000 times produced a distribution of performance changes that reflected the behavioral variability." It is not clear what 'sessions' means for "unsupervised optogenetic experiments". It is not clear why there are replacements from mice in the bootstrap method. Optogenetic experiment allow quantifying effect at the level of individuals: Test should be paired test and authors have to estimate individual distribution of performance changes under the null hypothesis. Finally, the authors used a unilateral test. This is fine, but the hypothesis should then be clearly stated. It is clearly expected that photo-stimulation of left S1 will reduce performance (with stimulation of the right whisker). The justification for a unilateral should be better justified for the other regions, and in particular the subcortical regions. The sentence l.560 "We next tested if the striatal optogenetic manipulation was sufficient to bias behavior" does not correspond to an unilateral test.

9. Finally, to obtain a proper control group, it is certainly better to use a control-AAV instead of no AAV. Could it be mentioned?

---

## [Author Response]

Essential Revisions:While the referees recognize the solidity of this work, a number of revisions are needed to better highlight its novelties with respect to previous publications on behavior during voluntary head-fixation, to reinforce the description of optogenetic silencing results and to help the reader grasp motivation aspects in this self-initiated behavior.

We thank the reviewers for the constructive comments. In the revised manuscripts, we have addressed these comments with the following revisions.

1. We strengthened the optogenetic manipulation results, including fraction of neurons affected and behavioral effect size relative to chance. We also quantitatively compared our effect size to previous studies and we discuss the interpretation of effect size.

2. We better highlight the advance provided by our method over previous methods and the utility of our workflow in surveying cortico-basal-ganglia loops. We discuss why this is currently difficult but is enabled by the advance here. We added new experiments probing additional sites in the striatum, further illustrating the ease and throughput of our workflow. Our mapping revealed a hotspot in the dorsolateral striatum that biased tactile-guided decision-making.

3. We added a new section in the Results examining behavioral signatures of motivation in the self-initiated behavior.

4. We have clarified method descriptions in various places.

5. We also added data from new optogenetic control mice in which GFP viruses were injected.

Please find our point-by-point response below.

1. Due to the authors' choice of plotting performance in optogenetics, it is not possible to know if the mice reach chance level during silencing. However, appreciating if the chance level is reached during silencing is important to evaluate if the silencing method leads to complete or partial impairments. The authors should thus also provide plots in which performance during optogenetics is express as the distance to chance level (e.g. 60% correct for a balanced binary task is 10% above chance level). Depending on the outcome of this re-plotting, the author should comment, based on the literature, whether incomplete effects are due to incomplete silencing or due to partial involvement of the target region in the task.

We have now added raw performance levels to Figure 7—figure supplements 1 and 2. In the power range tested in this study, photostimulation did not reduce performance to chance level (Figure 7—figure supplement 2). One limitation of the optogenetic workflow is the interpretation of behavioral deficit effect size. We examined this issue in ALM, a brain region from which we have the most extensive data (new Figure 7—figure supplement 1). In previous studies we have shown that bilateral photoinhibition of ALM results in chance level performance (Li et al. 2016, Figure 2b; Gao et al., 2018, Extended Data Figure 6b). Here, mice performance was above chance during photoinhibition of ALM (Figure 7—figure supplement 1). This difference in behavioral effect size likely resulted from incomplete silencing of ALM. The photostimulus intensity used here was much less than those used in previous studies (0.3 vs. 11.9 mW/mm^2^). In addition, a single virus injection was not sufficient to cover the entire ALM. Thus a partial behavioral effect could be due to incomplete silencing of a brain region, or partial involvement of the brain region in the task. Given this limitation, we caution that the function of a brain region could only be fully deduced in more detailed analysis and together with neurophysiology. We now better highlight these points in the Discussion (page 22-23).

Nevertheless, we believe our workflow is a highly useful tool to quickly screen brain regions that play roles in behavior.

1. At relatively low laser power, the effect sizes induced by photoinhibiting ALM and S1 are large and comparable to previous studies (Figure 7—figure supplement 1).

2. At relatively low laser power, we induced large and consistent biases by photostimulating subcortical regions that receive ALM and/or S1 inputs (dorsolateral striatum and SC).

3. We added new experiments showing that photostimulation in the posterior striatum (a region that does not receive ALM and S1 inputs) produced little effect. Thus, our method could differentiate brain regions involved in behavior from those that do not.

This makes more targeted analysis possible.

2. The comparison to other methods is important. As it stands, these seem marginal. This should be strengthened. It would be also important to highlight what types of questions, qualitatively, can be answered that are not possible (or difficult) otherwise.

The most significant advance of our workflow over previous methods is a substantial increase in the yield and duration of training. Our method also does not require pre-acclimation of mice and eliminates human supervision. We show that self-release is critical for automated training, which is missing from all current head-fixation systems for mice. Our automated training now rivals and slightly surpass the yield of manual training for the first time. We think this degree of automation is an important technical advance because it now enables systematic surveys of deep brain regions in behaviors that require thousands of trials to learn. We better highlight comparisons to previous methods in several key areas in the Supplemental Table 1. We have also strengthened the Discussion (page 20).

One line of inquiry immediately enabled by our automated training and optogenetic workflow is a systematic mapping of the cortico-basal-ganglia loops during difficult perceptual decision-making tasks. The striatum is topographically organized. Previous studies examined different subregions of the striatum in different perceptual decision behaviors, making comparisons across studies difficult. In the revised manuscript, we better highlight this issue using realistic calculations of yields and time. The striatum in the mouse brain is ~21.5 mm^3^ in size (Allen reference brain, (Wang, et al., Cell 2020)). Optogenetic experiments using optical fibers manipulate activity near the fiber tip (approximately 1 mm^3^). A systematic survey of different striatal domains’ involvement in specific behaviors is currently difficult. In our workflow, individual striatal subregions (~1 mm^3^, Figure 8) could be rapidly screened through parallel testing. At moderate throughput (15 mice / 2 months), a screen that tiles the entire striatum could be completed in under 12 months with little human effort. To illustrate its feasibility, we tested 3 subregions in the striatum previously implicated in different types of perceptual decision behaviors (Yartsev et al., *eLife* 2018; Sippy et al., Neuron 2015; Znamenskiy and Zador, Nature 2013), including an additional region in the posterior striatum that does not receive ALM and S1 inputs. The results revealed a hotspot in the dorsolateral striatum that biased tactile-guided decision-making (Figure 8). Our approach thus opens the door to rapid screening of the striatal domains during complex operant behaviors.

Moreover, by eliminating human intervention, automated training allows quantitative assaying of task learning (Figure 4). Home-cage testing also exposes behavioral signatures of motivation in self-initiated behavior (Figure 6). These observations suggest additional opportunities for inquires of goal-directed behaviors in the context of home-cage testing.

3. The task structure needs to be clarified. The sample period (1.3 sec) is followed by a delay epoch (1.3 sec), an auditory go cue and the response epoch. It is indicated that "Mice were free to respond by licking at any time during the trial, but only the first lick after the 'go' cue were registered as choice". The question is what exactly is a trial ? Is it: (i) a sequence sample-delay – response, with only the first lick taken as a response. That is, 1 head fixation = a trial and is associated with a maximum of 1 water reward (2-3 µL). (ii) a succession of sample-delay -1 response sequences during 30 or 60 sec. That is, 1 head fixation is associated with more than one water reward (2-3 µL). I suspect that the first proposal is the correct one, but this should be clearly stated in the manuscript. Along that line, Figure 3B is confusing. Authors should may be indicate the first lick, the one registered as choice in full color and the additional one as transparent?

In each head-fixation (30-60 s), mice were tested in a succession of trials. A sequence of sample-delay-response epochs constituted a trial (~5 s). Each trial was followed by an inter-trial-interval (2.5 s), after which the next trial began, until the head-fixation is released (Video 1). During the early phases of training (‘learn directional licking’ and ‘learn discrimination’), mice were free to lick any time during the trial. After the delay epoch was added, licking before the ‘go’ cue triggered a brief timeout, but mice were allowed to complete the trial. In all cases, the water reward was only contingent on the first lick after the ‘go’ cue. We have clarified this in the Results (page 9) and Methods (page 32). We have also revised Figure 3B as suggested by the reviewer to indicate the choice lick.

4. The authors should provide information about the daily water consumption (2-3 µL per lick number of correct trials per days) and whether it is stable during the protocol. Considering that a mouse drinks about 3-5 mL daily if it weighs more than 30 grams, it would correspond to 2000 correct trials more than 16 h of 30 sec head fixed session, if the above interpretation of a trial is correct.

We have added a new Figure 6 and a new section in the Results (page 13-14) describing detailed water consumption and body weight information in home-cage testing. At steady state, a mouse typically consumed ~1mL of water daily (~400 rewarded trials) while maintaining stable body weight. This amount of water consumption is similar to mice engaged in daily manual experiments (Guo et al., Plos ONE 2014).

5. In a paper (Torquet et al. 2018) which describes mouse behavior in an automated T-maze task (left or right to access water versus water + sugar), mice showed a decreased return time after choosing the less-rewarded side. This indicated an increased motivation after a failure, but also the fact that some trial can be associated with low motivation for the reward and engagement in the test just for exploration. Here, the authors showed a distribution of inter-fixation interval, with a long tail. Are these inter-fixation intervals correlated with the success or failure in the last trial which could indicate different motivation depending on the intervals?

We thank the reviewer for pointing us to this study, which was missed in our previous survey of the literature. We also now include citations to this line of work in the Introduction and Discussion (page 4 and 20).

We examined the inter-fixation-interval as suggested by the reviewer. Interestingly, the inter-fixation-interval after an error (which led to no reward) was significantly longer than following a correct trial (Figure 6E). This is inconsistent with error from exploration. Rather it likely reflects a loss of motivation after an error, perhaps due to the loss of an expected reward. In Torquet et al., mice showed a decreased return time after a less rewarding choice. Several factors could have contributed to this apparent difference. In Torquet et al., the location associated with higher reward was relatively constant (swapped every 3-4 days). A failed attempt in that context was likely due to exploration and a failed exploration would mean a higher reward is available on the other arm. Here, the context is different in that mice were learning to associate a sensory stimulus with a motor response. Licking either side could have caused an error depending on the sensory stimulus. When mice have not learned to associate the sensory stimulus with lick direction. We suspect that an error trial violated the mice’s expectation of reward, and therefore discouraged the mice, leading to a loss in motivation. Consistent with this interpretation, we also found a significant increase in inter-fixation-intervals shortly after a sensorimotor contingency reversal (Figure 6F). This coincided with an increase in error rate due to the rule change (Figure 5B). However, we recognize these are complex factors that would require further study to systematically dissect.

These new analyses have led us to include a new section in the revised manuscript (page 13-14 and Figure 6).

6. In the contingency reversals task, do the animals adapted their inter-fixation intervals just after the reversal?

Mice indeed increased their inter-fixation intervals just after the reversal (Figure 6F). This was likely due to failures to obtain expected rewards due to the rule change, leading to a loss of motivation. Inter-fixation interval recovered after mice learned the new contingency (Figure 6F). These results have been added to the new section of the manuscript (page 14).

7. The position of the pole with respect to the head (stimulation of the left or right whiskers) is not clearly indicated in the method, information seems to be shown only in Figure 3A. This should be indicated clearly. Along the same lines, on line 498, replace "Photoinhibition of S1" by "Photoinhibition of left S1".

We have added this information as suggested.

8. The statistics used in the optogenetic experiment is based on bootstrap and unilateral testing (one tail test). It is indicated that "In each round of bootstrap, we replaced the original behavioral dataset with a re-sampled dataset in which we re-sampled with replacement from: (1) mice, (2) sessions performed by each mouse, (3) trials within each session. We then computed the performance change on the re-sampled dataset. Repeating this procedure 10,000 times produced a distribution of performance changes that reflected the behavioral variability." It is not clear what 'sessions' means for "unsupervised optogenetic experiments". It is not clear why there are replacements from mice in the bootstrap method. Optogenetic experiment allow quantifying effect at the level of individuals: Test should be paired test and authors have to estimate individual distribution of performance changes under the null hypothesis. Finally, the authors used a unilateral test. This is fine, but the hypothesis should then be clearly stated. It is clearly expected that photo-stimulation of left S1 will reduce performance (with stimulation of the right whisker). The justification for a unilateral should be better justified for the other regions, and in particular the subcortical regions. The sentence l.560 "We next tested if the striatal optogenetic manipulation was sufficient to bias behavior" does not correspond to an unilateral test.

We thank the reviewer for catching these. We have clarified the description in the Methods (page 37). For home-cage optogenetic experiments, each day (dark + light cycle) was treated as a ‘session’.

We clarify that in each round of bootstrap, for each mouse we computed its performance change in photostimulation trials relative to the control trials, this is equivalent to a paired test (i.e. a change across conditions within-individuals). The resampling of mice was done before performance change was computed in each mouse. In essence, the *nested* bootstrap produced a distribution of observed performance change (within-individuals) taking into account variabilities across 3 levels: (1) mice included in the analysis, (2) sessions from each mouse, (3) trials from each session. The distribution was used to assess the probability of observing a behavioral change in the opposite direction from that observed (i.e. one-tailed test).

We now justify the use of unilateral photostimulation in the text (page 15 and 17-18). In our task, mice discriminate object location using the right whiskers and report choice using directional licking. We targeted the left barrel cortex, contralateral to the side of the tactile stimulus. For ALM, we tested each hemisphere because previous studies show that unilateral photoinhibition of ALM biases licking to the ipsilateral direction (Guo et al., Neuron 2014; Li et al., Nature 2015). This enabled comparisons to previous studies. For striatum and superior colliculus, previous studies suggest their roles in driving directional licking are lateralized (Lee and Sabatini, BioRxiv 2020; Duan, et al., BioRxiv 2019). In addition, the dorsolateral striatum targeted here receives inputs predominately from the ipsilateral barrel cortex (Sippy et al., Neuron 2015). We therefore targeted the left striatum, ipsilateral to the left barrel cortex and contralateral to the tactile stimulus.

9. Finally, to obtain a proper control group, it is certainly better to use a control-AAV instead of no AAV. Could it be mentioned?

We have tested a new group of GAD2-Cre mice in which AAV-pCAG-FLEX-EGFP-WPRE viruses were injected into the left ALM. Photostimulation induced no detectable effect on behavioral performance. This data now replaces the data from mice with no viruses injected (Figure 7—figure supplement 2A).